# Recruited macrophages that colonize the post-inflammatory peritoneal niche convert into functionally divergent resident cells

P. A. Louwe [1], L. Badiola Gomez [1], H. Webster[2], G. Perona-Wright [2], C. C. Bain [1], S. J. Forbes [3] & S. J. Jenkins [1]✉

Inflammation generally leads to recruitment of monocyte-derived macrophages. What regulates the fate of these cells and to what extent they can assume the identity and function of resident macrophages is unclear. Here, we show that macrophages elicited into the peritoneal cavity during mild inflammation persist long-term but are retained in an immature transitory state of differentiation due to the presence of enduring resident macrophages. By contrast, severe inflammation results in ablation of resident macrophages and a protracted phase wherein the cavity is incapable of sustaining a resident phenotype, yet ultimately elicited cells acquire a mature resident identity. These macrophages also have transcriptionally and functionally divergent features that result from inflammation-driven alterations to the peritoneal cavity micro-environment and, to a lesser extent, effects of origin and time-of-residency. Hence, rather than being predetermined, the fate of inflammation-elicited peritoneal macrophages seems to be regulated by the environment.

[1] Centre for Inflammation Research, Queens Medical Research Institute, Edinburgh EH16 4TJ, United Kingdom. [2] Institute of Infection, Immunity & Inflammation, University of Glasgow, Glasgow G12 8TA, United Kingdom. [3] Centre for Regenerative Medicine, University of Edinburgh, Edinburgh EH16 4UU, United Kingdom. ✉email: stephen.jenkins@ed.ac.uk

nflammation alters the composition and function of the tissue macrophage compartment, typically leading to substantial recruitment of monocytes from the blood and activation or even loss of the tissue-resident cells[1]. While these early cellular processes have been well-characterized across various tissues and models of inflammation, it remains unclear how homeostasis within the macrophage compartment is reinstated post inflammation and consequently what long-term effects inflammation may have on tissue macrophage function.

In the steady-state, resident macrophages across tissues share expression of core lineage-related genes upon which a tissue-specific transcriptional, epigenetic, and functional identity is overlaid[2–5]. These unique molecular identities are largely established upon exposure to tissue-specific environmental signals that in turn drive expression of tissue-specific transcription factors. Tissue resident macrophages also have diverse developmental origins[6,7] with many tissues containing self-renewing populations largely seeded during embryogenesis and short-lived bone marrow (BM)-derived populations that seemingly inhabit distinct anatomical regions[8,9]. However, in the absence of embryonically seeded macrophages, circulating monocytes appear able to give rise to long-lived and transcriptionally and functionally normal resident cells[10–14], suggesting origin may not be a key determinant of macrophage identity per se, but rather tissue-specific anatomically restricted signals and cell interactions constitute a niche that controls macrophage longevity and gene expression. While a small number of seemingly ontogeny-related transcriptional differences may distinguish embryonic and monocyte-derived resident macrophages present within the same niche[12,14], limited evidence suggests that even these may be gradually reprogrammed over time[13,15]. Based on these observations, it has been proposed that competition for signals and cell-to-cell interactions that drive survival and proliferation of macrophages and expression of tissue-specific transcription factors dictates the balance between incumbent resident macrophages and infiltrating monocytes[16,17].

Macrophages in the peritoneal cavity regulate peritoneal B1 cells[15,18] and provide immune surveillance of the cavity[19] and neighboring tissues[20] but they are also implicated in many pathologies, including endometriosis, post-surgical adhesions, pancreatitis, and metastatic cancer[19]. The peritoneal cavity contains two populations of resident macrophages: an abundant population of so-called "large" peritoneal macrophages (LPM) that are embryonically seeded and long-lived, and a rarer population of short-lived MHCII$^+$ monocyte-derived cells termed small peritoneal macrophages (SPM)[21]. The transcriptional identity of LPM is dependent upon the transcription factors GATA6[5,22,23] and CEBPβ[24] while SPM depend upon IRF4[25]. Expression of GATA6 by LPM is driven primarily by omentum-derived retinoic acid[5] at least in part via retinoid X receptors alpha and beta (RXRα/β)[26]. Despite initially having an embryonic origin, the LPM population is gradually replaced by monocyte-derived cells with age, a process that occurs more rapidly in males[21]. Thus, differential rates of replenishment alters the time-of-residency of each macrophage, which leads to differences in phenotype and function of LPM between the sexes and with age[15,21]. Indeed, single-cell RNA-sequencing of peritoneal macrophages has revealed LPM comprise multiple transcriptionally distinct subsets[15,27] that, at least in females, appear to represent states associated with different times-of-residency[15]. However, while tissue-resident macrophages in solid organs are envisaged to have a static architectural niche comprising stable cell interactions[16,28] as delineated in several tissues[29–31], peritoneal macrophages "float" in a fluidic environment[32] implying more complex interactions control their maintenance, identity and sub-specialization.

Monocyte-derived macrophages recruited during inflammation typically exhibit distinct transcriptional, functional and phenotypic signatures to resident cells, even in tissue sites where resident cells are ordinarily replenished by monocytes[33,34]. However, it's unclear whether inflammatory macrophages are fully capable of reprogramming to become resident macrophages and hence if their fate is predominantly regulated by access to appropriate niche signals or rather predetermined during their initial differentiation. Macrophages recruited to the alveolar space following influenza infection or bleomycin-induced lung damage persist for many months[35,36], but retain significant transcriptional differences to enduring resident cells. Notably, established resident macrophages exhibit a relatively poor ability to engraft and reprogram upon adoptive transfer into an ectopic tissue site[4,14], suggesting that differentiation of macrophages may lead to substantial loss in plasticity. Irrespective, if reprogramming of inflammatory macrophages also has an element of time-dependence, their persistence would be predicted to lead to prolonged alteration in the functional capacity of the tissue macrophage compartment.

In the peritoneal cavity, sterile inflammation can cause substantial contraction in number of LPM through cell death or loss in fibrin clots[32,37] but the extent of this loss appears dependent on stimulus and severity of inflammation[37,38]. While remaining LPM can subsequently proliferate during the resolution phase[39], peritoneal inflammation[40,41], including that caused by abdominal surgery[15] can lead to at least partial replacement of the long-lived LPM population from the BM, with the degree of replacement seeming to correlate with the extent of the preceding loss of incumbent cells[37]. The functional implications of displacement of the resident population remains unclear.

Here, we study the peritoneal cavity to investigate what regulates the fate of inflammatory macrophages following sterile inflammation. Using adoptive transfer to unequivocally track inflammatory macrophages and identify the degree to which the environment controls the fate of these cells, we demonstrate that macrophages infiltrating the cavity after mild inflammation persist long-term, but that competition with incumbent resident macrophages inhibits effective acquisition of a mature resident phenotype. Consistent with this competition model, severe inflammation, which causes ablation of incumbent resident macrophages, results in conversion of inflammatory macrophages to mature resident cells. We therefore identify the existence of a biochemical niche for resident peritoneal macrophages. Competition for this niche controls the capacity of monocyte-derived cells to undergo conversion to mature resident macrophages and a failure to compete retains them in a highly proliferative and immunoregulatory state.

## Results

**Inflammatory macrophages persist after mild peritoneal inflammation.** To investigate what regulates the fate and phenotype of inflammatory and resident macrophages following resolution of sterile peritoneal inflammation, we used a well-characterized model of intraperitoneal injection with low-dose zymosan A (10 μg/mouse), in which both populations remain present following resolution of the neutrophilic phase[38,39,42]. First, to definitively delineate incumbent resident cells from inflammatory macrophages recruited during the acute phase of inflammation, we utilized an established method of injecting fluorescent PKH26-PCL dye intraperitoneally 24 h before zymosan to exclusively label peritoneal phagocytes present prior to inflammation[40]. Uptake of PKH26-PCL dye was largely restricted to all resident LPM and most SPM (Supplementary Fig. 1a), identified as F4/80$^{Hi}$ or F4/80$^{Lo}$ CD226$^+$ cells, respectively[25,42],

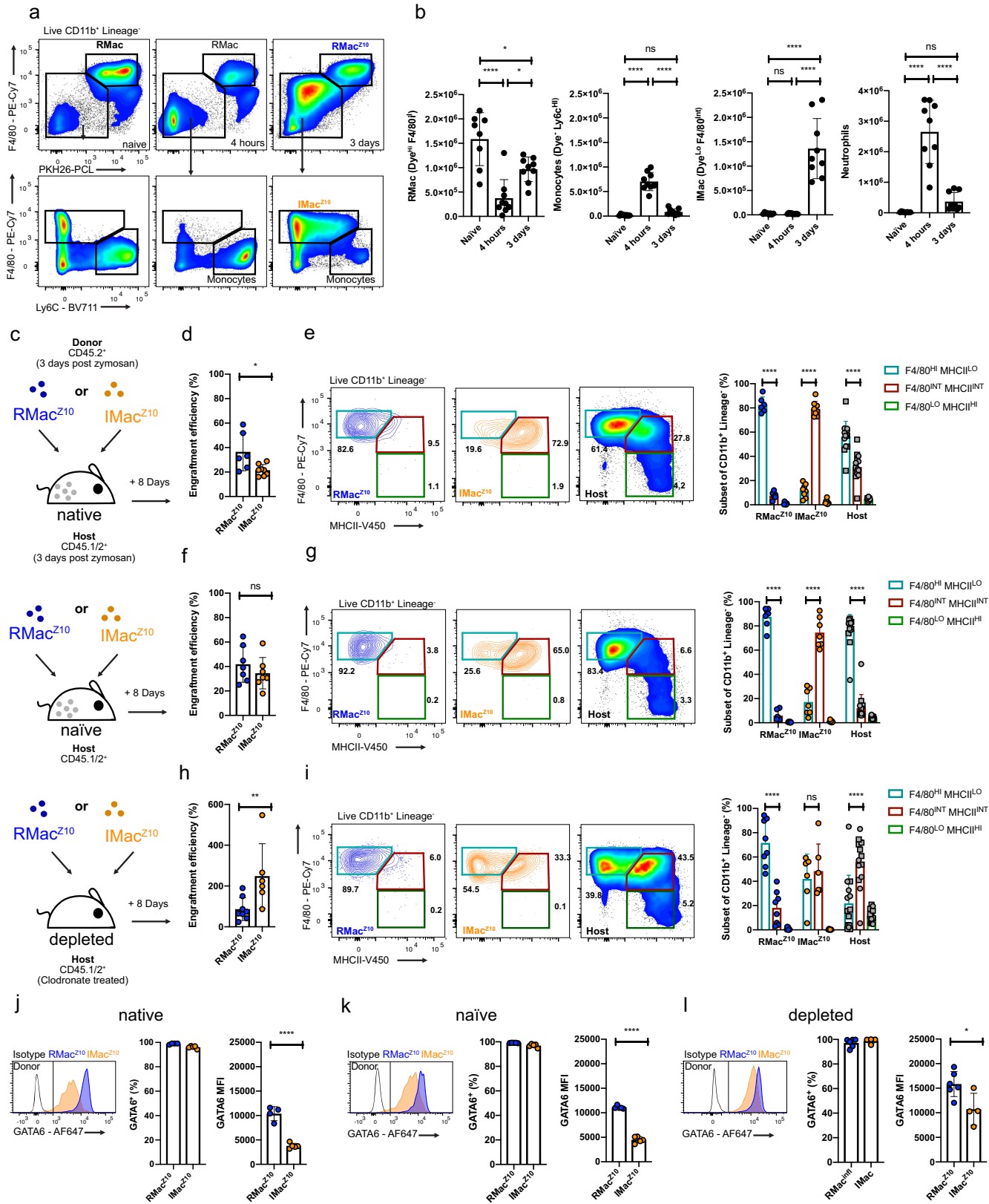

and no free dye remained 24 h later (Supplementary Fig. 1b, c). Subsequent injection of low-dose zymosan-induced disappearance of dye-labeled (Dye$^{Hi}$) F4/80$^{Hi}$ resident macrophages and influx of dye-negative (Dye$^{Lo}$) Ly6c$^+$ monocytes and neutrophils within 4 h. By day 3, neutrophils were largely cleared and remaining Dye$^{Lo}$ infiltrating cells now exhibited a predominantly F4/80$^{Int}$ Ly6c$^{Lo/Int}$ phenotype consistent with inflammatory macrophages[33,38] and dye$^{Hi}$ F4/80$^{Hi}$ resident macrophages had partially recovered in number (Fig. 1a, b), consistent with their reported repopulation by self-renewal in this model[39]. Finally, to validate our dye-based tracking system, we assessed dye-labeling in tissue-protected BM chimeric mice, which allow recruited and

**Fig. 1 Competition mediates inflammatory macrophage phenotype early post resolution. a** F4/80$^{HI}$ PKH26-PCL$^{Hi}$ resident macrophages, PKH26-PCL$^{Lo}$ Ly6c$^+$ monocytes and PKH26-PCL$^{Lo}$ F4/80$^{Int}$ inflammatory macrophages in the naïve peritoneal cavity, 4 h and 3 days post 10 μg zymosan. **b** Number of RMac, Monocytes, IMac, and Neutrophils the naïve peritoneal cavity ($n = 8$), 4 h post zymosan ($n = 9$) and 3 days post zymosan ($n = 9$). *$p < 0.05$, ****$p < 0.0001$, one-way ANOVA with Tukey's multiple comparisons test. **c** Experimental scheme for transfer of RMac$^{Z10}$ (blue) or IMac$^{Z10}$ (orange) into mirroring inflamed (native), naïve or depleted recipient mice. **d** Engraftment efficiency of RMac$^{Z10}$ ($n = 6$) and IMac$^{Z10}$ ($n = 8$) 8 days after transfer into inflamed recipients. $p = 0.029$(*), Mann–Whitney test. **e** F4/80 and MHCII expression by RMac$^{Z10}$ ($n = 6$), IMac$^{Z10}$ ($n = 8$) or host ($n = 14$) cells 8 days post transfer. Each comparison $p < 0.0001$(****), determined by two-way ANOVA and post hoc Tukey test. **f** Engraftment efficiency of RMac$^{Z10}$ ($n = 7$) and IMac$^{Z10}$ ($n = 7$) 8 days after transfer into naïve recipients. Significance determined using Mann–Whitney test. **g** F4/80 and MHCII expression by RMac$^{Z10}$ $n = 7$), IMac$^{Z10}$ ($n = 7$), or host ($n = 14$) cells 8 days after transfer. Each comparison $p < 0.0001$(****), two-way ANOVA and post hoc Tukey test. **h** Engraftment efficiency of RMac$^{Z10}$ ($n = 8$) and IMac$^{Z10}$ ($n = 6$) 8 days after transfer into clodronate-depleted recipients. $p = 0.008$(**), Mann–Whitney test. **i** F4/80 and MHCII expression by RMac$^{Z10}$ ($n = 8$), IMac$^{Z10}$ ($n = 6$), or host ($n = 14$) cells 8 days after transfer. Each comparison $p < 0.0001$(****), two-way ANOVA and post hoc Tukey test. **j** Proportion of RMac$^{Z10}$($n = 4$), IMac$^{Z10}$ ($n = 5$) that are GATA6$^+$ and MFI of GATA6 expression 8 days after transfer into inflamed recipients. $p < 0.0001$ (****), student's $t$-test. **k** Proportion of RMac$^{Z10}$ ($n = 5$) and IMac$^{Z10}$ ($n = 5$) that are GATA6$^+$ and MFI of GATA6 expression 8 days after transfer into naïve recipients. $p < 0.0001$ (****), student's $t$-test. **l** Proportion of RMac$^{Z10}$ ($n = 6$), IMac$^{Z10}$ ($n = 5$) that are GATA6$^+$ and MFI of GATA6 expression 8 days after transfer into clodronate-depleted recipients. $p = 0.023$(*), student's $t$-test. Experiments are presented as mean ± standard deviation. Each symbol represents an individual animal. Data were pooled from at least two independent experiments. Host cells represented by squares or circles are from recipients of RMac$^{Z10}$ or IMac$^{Z10}$, respectively.

resident cells to be determined definitively[21]. This confirmed that Dye$^{Lo}$ F4/80$^{Int}$ macrophages present in the peritoneal cavity at day 3 were derived from recruited cells as evidenced by their high levels of non-host chimerism, whereas Dye$^{Hi}$ F4/80$^{Hi}$ cells displayed low levels of chimerism, demonstrating their tissue residency (Supplementary Fig. 1d). Moreover, Dye$^{Lo}$ F4/80$^{Int}$ had high levels of MHCII and virtually no expression of the resident macrophage marker Tim4 (Supplementary Fig. 1e), features known to differentiate inflammatory from resident macrophages during resolution[33,38]. Thus, PKH26-PCL-labeling faithfully delineated resident vs recruited macrophage subsets and importantly, this system used a minimal number of surface antibodies thereby circumventing potential confounding effects of adoptive transfer of antibody-coated cells.

Next, we used adoptive transfer to unequivocally determine the fate of these populations. Specifically, Dye$^{Hi}$ F4/80$^{Hi}$ resident macrophages (RMac$^{Z10}$) and Dye$^{Lo}$ F4/80$^{Int}$ inflammatory macrophages (IMac$^{Z10}$) were FACS-purified from C57BL/6 WT (CD45.2$^+$) donor mice 3 days after injection of low-dose zymosan (Fig. 1a) and transferred intraperitoneally into separate congenic WT (CD45.1/2$^+$) host animals. Recipient mice had been pre-treated 3 days prior with an equivalent dose of zymosan to ensure labeled cells were transferred into a similar environment (Fig. 1c, native). Eight days post transfer, transferred donor RMac$^{Z10}$ and IMac$^{Z10}$ exhibited a similar degree of engraftment, defined as the number retrieved as a proportion of those transferred, although this was somewhat greater for RMac$^{Z10}$ (Fig. 1d). Whereas transferred RMac$^{Z10}$ remained predominantly MHCII$^{Lo}$, IMac$^{Z10}$ remained largely MHCII$^{Hi}$ and continued to express marginally less F4/80 such that the two donor populations were identified with relative accuracy using these markers (Fig. 1e). Critically, virtually all transferred IMac$^{Z10}$ expressed the LPM-specific transcription factor GATA6 but at markedly lower levels than RMac$^{Z10}$ (Fig. 1j). The host CD11b$^+$ myeloid compartment also contained a mixture of F4/80$^{Int/Hi}$ MHCII$^{Hi}$ GATA6$^+$ and F4/80$^{Hi}$ MHCII$^{Lo}$ GATA6$^+$ macrophages, consistent with persistence of endogenous inflammatory and resident macrophages, but also a minor fraction of F4/80$^{Lo}$MHCII$^{Hi}$GATA6$^-$ cells (Fig. 1e and Supplementary Fig. 2a) suggestive of newly generated SPM and/or CD11b$^+$ DCs. Hence, by combining dye-labeling and adoptive transfer, we have developed a robust system to identify and fate map tissue-resident and inflammatory macrophages in the context of peritoneal inflammation and reveal that distinct populations of MHCII$^-$ and MHCII$^+$

peritoneal macrophages persist following zymosan-induced peritoneal inflammation due to endurance of tissue-resident macrophages established prior to inflammation and monocyte-derived macrophages recruited at the onset of inflammation, respectively.

**Resident macrophages limit initial survival and phenotype of IMac$^{Z10}$.** We next explored what regulates the short-term fate of these cells. First, we transferred RMac$^{Z10}$ and IMac$^{Z10}$ into naïve-recipient mice (Fig. 1c, naïve) to determine whether their survival and phenotype is dictated primarily by the post inflammation micro-environment. In this non-inflamed environment both donor populations persisted equally (Fig. 1f), with a level of engraftment akin to that observed for RMac$^{Z10}$ transferred to inflamed mice (Fig. 1d). Despite this, IMac$^{Z10}$ remained MHCII$^{Hi}$ (Fig. 1g) and expressed intermediate levels of GATA6 (Fig. 1k), suggesting this phenotype was not a product of the post-inflammatory micro-environment.

Next to determine if competition with resident macrophages regulates survival and phenotype of IMac$^{Z10}$, we pre-treated recipient mice 7 days prior to transfer with clodronate-loaded liposomes (Fig. 1c, depleted). This regime caused rapid and prolonged loss of recipient F4/80$^{Hi}$ LPM, with the cavity being essentially devoid of these cells at the point of adoptive transfer at day 7 (Supplementary Fig. 2b). In the absence of endogenous resident macrophages, engraftment efficiency of IMac$^{Z10}$ was approximately 250%, indicating these cells have the ability to expand to fill the empty niche (Fig. 1h). Furthermore, nearly 50% of IMac$^{Z10}$ adopted a more resident-like MHCII$^{Lo}$ phenotype (Fig. 1i) but failed to acquire similar levels of GATA6 as RMac$^{Z10}$ within this period (Fig. 1l). Surprisingly, although RMac$^{Z10}$ also persisted better in the depleted environment, with an engraftment efficiency nearer 100%, they were unable to expand to the same degree as IMac$^{Z10}$ (Fig. 1h). Notably, host macrophages also repopulated the cavity during this period, yet they largely exhibited an MHCII$^{Hi}$ phenotype (Fig. 1i) resembling that of IMac$^{Z10}$ in their native inflamed environment, suggesting these cells likely derive from Ly6C$^+$ monocytes recruited to the cavity post-depletion (Supplementary Fig. 2b). We also found that irrespective of environment, nearly all IMac$^{Z10}$ expressed the GATA6-independent LPM marker CD102[5] yet few expressed Tim4 (Supplementary Fig. 2c). Altogether, these data suggest that while IMac$^{Z10}$ persist through the early phases of resolution, their survival and conversion to MHCII$^{Lo}$ cells is largely regulated by the presence of competing resident macrophages.

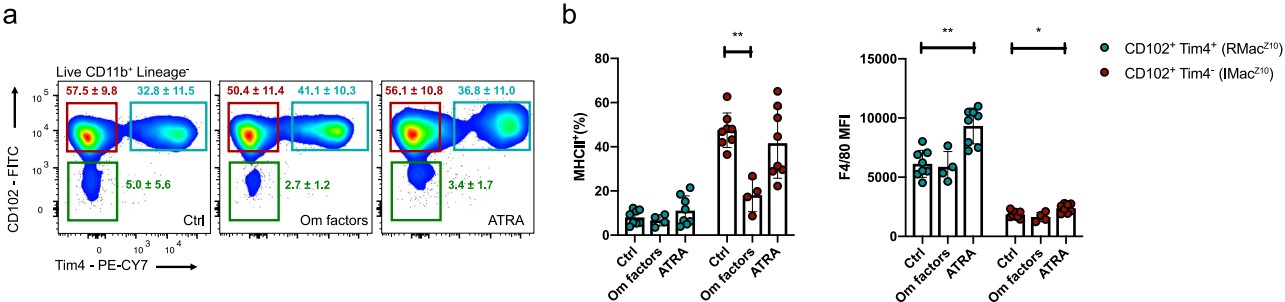

**Fig. 2 Inflammatory macrophages are responsive to niche factors in vitro. a** Expression of CD102 and Tim4 by cultured cells after 24 h culture with indicated treatment. **b** Proportion of macrophage subsets that express MHCII and F4/80 MFI after 24 h culture with control medium ($n = 8$), Omentum factor containing medium ($n = 4$) or ATRA containing medium ($n = 8$). MHCII$^+$: $p = 0.0046$ (**), F4/80 MFI: $p = 0.0028$ (**), $p = 0.032$ (*), determined by one-way ANOVA and Dunnet's multiple comparisons test for each subset individually, followed by Bonferroni adjustment. Data was pooled from 2 independent experiments and are presented as mean ± standard deviation. For each treatment group a symbol represents a culture well derived from an individual mouse.

GATA6 expression by LPM is largely induced by retinoic acid from omental and peritoneal stromal cells whereas the omentum produces additional factors that can drive retinoic acid-independent features of LPM[5,43]. We therefore cultured peritoneal cells collected 11 days post zymosan with all trans retinoic acid (ATRA) or omentum culture supernatant (Om factors) for 24 h to determine whether MHCII expression by inflammatory macrophages is responsive to retinoic acid or other omental factors. As CD102 and Tim4 did not appear to be altered in any of the in vivo experiments we used these to identify resident (CD102$^+$/Tim4$^+$) and inflammatory (CD102$^+$Tim4$^-$) macrophages post culture. Indeed, post culture expression of these surface markers remained unchanged between treatments (Fig. 2a). Culture with ATRA led to increased expression of the GATA6 responsive marker F4/80[5] by CD102$^+$/Tim4$^+$ resident and CD102$^+$Tim4$^-$inflammatory macrophages but not downregulation of their MHCII expression. In contrast, culture with omental supernatant led to downregulation of MHCII by CD102$^+$Tim4$^-$inflammatory macrophages (Fig. 2b). Hence, the presence of competing resident macrophages may limit the differentiation of inflammatory macrophages to a MHCII$^{Lo}$ resident phenotype by restricting availability of retinoic acid-independent signals from the omental niche.

Subsequently, to ascertain whether access to retinoic acid was the limiting factor in the degree of GATA6 expression by IMac$^{Z10}$ in vivo, we administered ATRA or oil vehicle every other day following adoptive transfer of RMac$^{Z10}$ or IMac$^{Z10}$ into inflamed recipient mice (Supplementary Fig. 2d). Unfortunately, injection of the oil vehicle led to complete replacement of the macrophage compartment by host Ly6C$^+$ monocytes and F4/80$^{lo}$ macrophages as no donor cells were detected at day 8 in any group (Supplementary Fig. 2e). However, while only a minor fraction of host F4/80$^{lo}$ macrophages expressed high levels of GATA6, treatment with ATRA increased this almost 4-fold (Supplementary Fig. 2f, g), suggesting the availability of retinoic acid limits GATA6 expression by inflammatory macrophages.

**IMac$^{Z10}$ retain intrinsic and environment-driven features long-term.** To investigate whether IMac$^{Z10}$ can persist long-term and assimilate into the resident LPM compartment, we continued to track these and the prevailing RMac$^{Z10}$ following transfer into native inflamed cavities versus macrophage-depleted cavities and assessed their phenotype after 8 weeks. To understand whether inflammation changed the behavior and long-term fate of RMac$^{Z10}$, we included F4/80$^{Hi}$ Dye$^{Hi}$ LPM from naïve donors (RMac) transferred into naïve mice or macrophage-depleted animals for comparison (Fig. 3a). Notably, only 60% of the

transferred RMac$^{Z10}$ retained PKH26-PCL-labeling by this time while some recipient cells had acquired dye (Supplementary Fig. 3a) confirming the need for adoptive transfer to accurately discriminate these cells. In these experiments, a similar proportion of transferred IMac$^{Z10}$ persisted in their native environment to both RMac$^{Z10}$ and RMac (Fig. 3b, left). However, retrospective pooling of all data generated from this time-point throughout our study (Figs. 3b with 5b) revealed an overall pattern that was similar to day 8, whereby IMac$^{Z10}$ persisted marginally less well than their RMac$^{Z10}$ counterparts (Supplementary Fig. 3b). Indeed, the overall similarity in survival of donor IMac$^{Z10}$ and RMac$^{Z10}$ between day 8 (Fig. 1d) and week 8 (Fig. 3b) post transfer suggests little loss of either population occurred in this time and demonstrates that macrophages elicited by an inflammatory agent become long-lived resident macrophages. Furthermore, the comparable survival of both IMac$^{Z10}$ and RMac$^{Z10}$ to RMac in naïve mice suggests that as early as day 3 post zymosan injection the homeostatic mechanisms regulating longevity/autonomy of peritoneal macrophages are reinstated. Following transfer into depleted recipients, IMac$^{Z10}$ again expanded significantly in number whereas RMac$^{Z10}$ did not (Fig. 3b, right). Likewise, the similarity in persistence of engrafted IMac$^{Z10}$ and RMac$^{Z10}$ between day 8 (Fig. 1h) and week 8 post transfer (Fig. 3b) suggests that the resident peritoneal macrophage pool also quickly re-establishes following depletion and resumes self-maintenance irrespective of origin.

Importantly, even after 8 weeks in their native environment IMac$^{Z10}$ exhibited lower expression of GATA6, marginally less F4/80, and a higher proportion of MHCII$^+$ cells than either RMac population (Fig. 3d, e and Supplementary Fig. 3d). In contrast, prior removal of competing endogenous cells through administration of clodronate liposomes allowed transferred IMac$^{Z10}$ to fully acquire the MHCII$^{Lo}$GATA6$^{Hi}$ phenotype of RMac$^{Z10}$ within 8 weeks (Fig. 3d, e and Supplementary Fig. 3d). To investigate the wider transcriptional integration of IMac$^{Z10}$ within the resident LPM compartment, we sorted the 3 donor populations from both native and depleted environments at 8 weeks post transfer and investigated gene expression using the NanoString nCounter mouse myeloid panel. This analysis revealed that in their native environment IMac$^{Z10}$ remained highly transcriptionally distinct from RMac$^{Z10}$, with 78 of the 372 detected genes being differentially expressed (Supplementary Data 1, Fig. 3f, Supplementary Fig. 3e), whereas no detectable differences were apparent between RMac and RMac$^{10}$ (Supplementary Data 2). Of the 13 genes included in the panel that are unique to peritoneal LPM[5,43], 11 were differentially expressed between IMac$^{Z10}$ and RMac$^{Z10}$ (Supplementary Fig. 3f) including

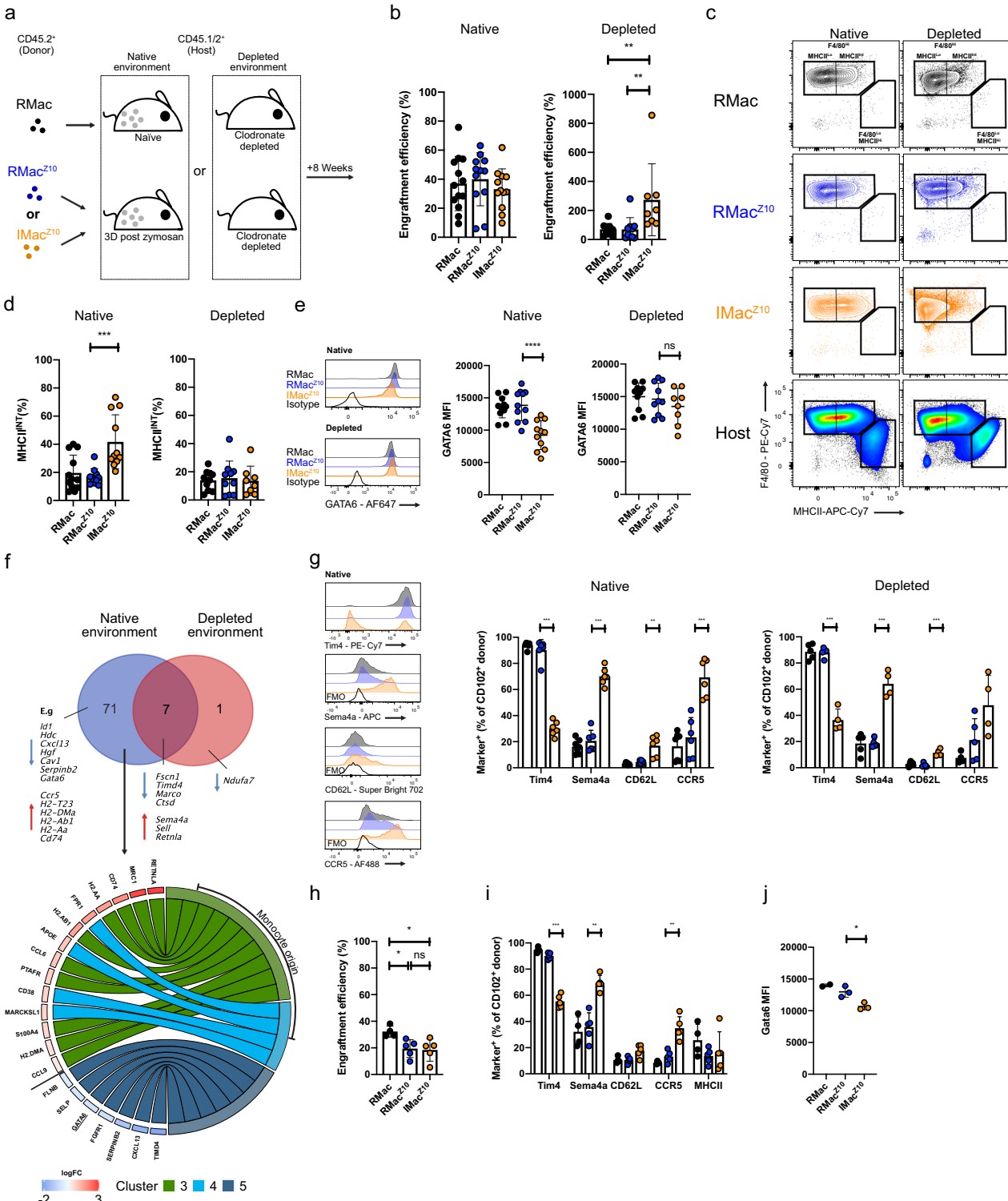

*Gata6*. In contrast, *Cebpb*, which encodes the transcription factor CEBPβ upon which LPM are also dependent[24], was expressed equally by IMac[Z10] (Supplementary Data 1). A quarter of the genes differentially expressed between IMac[Z10] and RMac[Z10] overlapped with those regulated by GATA6 in LPM[5,22,23], including *Adgre1*, which encodes F4/80, and consequently the gene signature of IMac[Z10] largely aligned with that of *Gata6*-deficient LPM[5,22,23] (Supplementary Fig. 3g). As GATA6 expression by LPM is regulated by retinoic acid through RXRα and RXRβ[26], we also assessed overlap with other genes regulated

by RXRAB independent of GATA6 (Supplementary Fig. 3g) and found that a further fifth of the genes that differentiated IMac[Z10] from RMac[Z10] to potentially arise from this pathway.

Furthermore, comparison with our published single-cell RNAseq analysis[15] of LPM revealed that genes expressed more highly in IMac[Z10] overlapped exclusively with those expressed more highly by LPM of recent monocyte origin in naïve female mice (Cluster 3 and 4, e.g. *Apoe*, *Retnla*, and genes related to MHCII presentation), whereas genes expressed more highly in RMac[Z10] overlapped exclusively with those expressed more

**Fig. 3 Long-lived colonizing inflammatory macrophages retain intrinsic and environment-dependent differences to RMac. a** Experimental scheme for transfer of RMac, RMac$^{Z10}$ or IMac$^{Z10}$ into naïve, inflamed or clodronate-depleted recipients. **b** Engraftment efficiency of RMac, RMac$^{Z10}$ and IMac$^{Z10}$ 8 weeks after transfer into native (left; $n = 13$, $n = 9$, $n = 9$) or depleted recipients (right; $n = 10$, $n = 10$, $n = 8$) **$p < 0.01$, one-way ANOVA and Tukey's multiple comparisons test. **c** Expression of F4/80 and MHCII 8 weeks after transfer into native or clodronate-depleted recipients. **d** Proportion of RMac, RMac$^{Z10}$, and IMac$^{Z10}$ that express MHCII 8 weeks after transfer into native (left; $n = 13,11,11$) or depleted recipients (right; 10, 10, 8). $p = 0.00037$ (***), one-way ANOVA and Tukey's multiple comparisons test. **e** GATA6 MFI 8 weeks after transfer of RMac, RMac$^{Z10}$, or IMac$^{Z10}$ into native (left; $n = 12,11,11$) or depleted (right; $n = 10,10,8$) recipients. $p < 0.0001$(****), one-way ANOVA and Tukey's multiple comparisons test. **f** Venn diagram indicating overlap between genes differentially expressed between RMac$^{Z10}$ and IMac$^{Z10}$ (adj $p$-value < 0.05), 8 weeks after transfer into native (blue) or depleted (red) recipients, and circus plot depicting differentially expressed genes that are cluster markers identified by Bain et al.[15]. **g** Marker expression by CD102$^+$ RMac (black), RMac$^{Z10}$ (blue) and IMac$^{Z10}$ (orange) 8 week after transfer into native (left; $n = 7,6,6$) or depleted recipients (right; $n = 5, 5, 4$). **$p < 0.01$, **$p < 0.01$ ***$p < 0.001$, one-way ANOVA and Dunnet's multiple comparisons test for each marker, followed by Bonferroni adjustment. **h** Engraftment efficiency of RMac, RMac$^{Z10}$, and IMac$^{Z10}$ 22 weeks after transfer into native recipients ($n = 4, 5, 5$). *$p < 0.05$, one-way ANOVA and Tukey's multiple comparisons test. **i** Marker expression by CD102$^+$ RMac (black), RMac$^{Z10}$ (blue) and IMac$^{Z10}$ (orange) 22 weeks after transfer into native ($n = 4, 5, 5$) recipients. **$p < 0.01$ ***$p < 0.001$, one-way ANOVA and Dunnet's multiple comparisons test for each marker, followed by Bonferroni adjustment. **j** GATA6 MFI, 22 weeks after transfer of RMac, RMac$^{Z10}$, or IMac$^{Z10}$ into native ($n = 2, 3, 3$) recipients. $p = 0.019$ (*), one-way ANOVA and Tukey's multiple comparisons test. Data are presented as mean ± standard deviation with each symbol representing an individual animal. Data were pooled from at least two independent experiments, except for **j**, which is a single experiment. Host cells represented by squares or circles are recipients of RMac$^{Z10}$ or IMac$^{Z10}$, respectively.

highly by the most long-lived LPM (Cluster 5, e.g. *Timd4*, *Cxcl13*, and *Gata6*) (Fig. 3f). Moreover, re-analysis of our single-cell RNA-seq dataset of LPM[15] revealed that cluster markers that define monocyte-derived LPM overlapped markedly and exclusively with genes expressed more highly by *Gata6*-deficient or *RXRAB*-deficient macrophages[5,22,23], whereas cluster markers of established LPM overlapped substantially and exclusively with genes expressed more highly by *Gata6*-sufficient or RXRAB-sufficient LPM (Supplementary Fig. 3h).

Hence, these data suggest that differences in retinoic acid signaling, either directly or via GATA6, between established resident macrophages and incoming monocyte-derived macrophages controls a significant proportion of the genes differentially expressed between these populations in steady-state and post inflammation. Critically, gene expression profiling suggested that IMac$^{Z10}$ and RMac$^{Z10}$ became transcriptionally more similar after transfer into depleted recipients, with the number of differentially expressed genes decreasing from 78 to 8 (Fig. 3f and Supplementary Data 3). Notably, differences in *Gata6* and almost all the potentially GATA6-regulated genes were lost (Supplementary Fig. 3i), as were differences in most other genes that defined clusters identified in steady-state[15], including *Cxcl13* and MHCII-related genes. Unsurprisingly, RMac and RMac$^{Z10}$ remained transcriptionally indistinct in the macrophage-deplete environment (Supplementary Data 4). Importantly, gene-ranking analysis indicated that neither RMac nor RMac$^{Z10}$ altered their transcriptional identity following transfer into depleted recipients (Supplementary Fig. 3j and Supplementary Data 5). This demonstrates that the transcriptional coalescence of IMac$^{Z10}$ and RMac$^{Z10}$ that occurs upon transfer into the macrophage-depleted environment is largely caused by transformation of IMac$^{Z10}$ into RMac$^{Z10}$ rather than the transformation of RMac$^{Z10}$ into IMac$^{Z10}$. Furthermore, these data suggest the majority of transcriptional differences between IMac$^{Z10}$ and RMac$^{Z10}$ are determined by the post-inflammatory environment or competition with incumbent resident macrophages for access to niche signals, whereas a smaller number of differentially expressed genes may represent cell-intrinsic features related to origin. Specifically, enduring resident macrophages seemingly prevent IMac$^{Z10}$ transition to a mature GATA6$^{hi}$ phenotype, thus retaining them in a transcriptional state associated with steady-state monocyte-derived LPM.

Flow-cytometric analysis confirmed that within the native post-inflammatory environment, IMac$^{Z10}$ expressed higher levels of Sema4a, CD62L, and CCR5 and but largely failed to acquire

expression of Tim4 (Fig. 3g, left), consistent with the differential expression of *Sema4a*, *Sell* (encoding for CD62L), *Ccr5*, and *Timd4* detected by NanoString. Similarly, IMac$^{Z10}$ in the depleted environment retained equivalently high levels of Sema4a and CD62L, and low levels of Tim4 expression (Fig. 3g, right), consistent with these being cell-intrinsic rather than environment-dependent features of IMac$^{Z10}$ (Fig. 3f). In line with an expression pattern predominantly dictated by environmental cues, we found that IMac$^{Z10}$ expressed more variable and on the whole lower levels of CCR5 in the macrophage-depleted environment (Fig. 3g, right). We extended this analysis to include surface markers that define newly monocyte-derived (Folate receptor β (FRβ)) and long-lived resident peritoneal macrophages (CD209b and V-set immunoglobulin domain-containing 4 (VSIG4))[15]. IMac$^{Z10}$ failed to acquire equivalent expression of CD209b or VSIG4 to either RMac$^{Z10}$ population irrespective of environment, whereas they exhibited comparatively high levels of FRβ in the native environment that, like CCR5, was lost in macrophage-depleted recipients, suggesting downregulation by environmental cues (Supplementary Fig. 3k). Finally, we determined whether reprogramming of IMac$^{Z10}$ may occur naturally over the lifespan following inflammation. Fate-mapping for 5 months revealed continued persistence of IMac$^{Z10}$, RMac$^{Z10}$ and RMac transferred into their native environments (Fig. 3h), although only RMac appeared to survive as well as at 8 weeks (Fig. 3b). Within this time IMac$^{Z10}$ had downregulated MHCII to levels equivalent to RMac$^{Z10}$, yet they continued to express lower levels of GATA6 and retain higher proportions of cells expressing Sema4a, CD62L, CCR5 (Fig. 3i, j) and FRβ (Supplementary Fig. 3l). Despite this, fewer IMac$^{Z10}$ expressed CCR5 or FRβ than at 8 weeks (Figs. 3g "native" vs. 3i and Supplementary Fig. 3k "native" vs. 3l), consistent with gradual reprogramming of expression of these markers, whereas expression of Sema4a and CD62L remained unchanged. Furthermore, IMac$^{Z10}$ had acquired equivalent levels of VSIG4 to RMac$^{Z10}$ by this time but failed to upregulate expression of Tim4 and CD209b to levels observed on the resident populations (Fig. 3i and Supplementary Fig. 3l), despite the frequency of IMac$^{Z10}$ expressing these markers increasing compared to 8 weeks (Fig. 3g "native" vs. 3i and Supplementary Fig. 3k "native" vs. 3l). Of note, the low frequency of RMac and RMac$^{Z10}$ that expressed CCR5 and FRβ by week 8 of transfer was seemingly reduced even further by 5 months, while the proportion that expressed Vsig4 and CD209b continued to rise gradually (Fig. 3g "native" vs. 3i and Supplementary Fig. 3k "native" vs. Supplementary Fig. 3l). These data are consistent with our previous supposition that expression of Tim4, CD209b and VSIG4

by LPM is regulated by time-of-residency and demonstrate this remains so following mild inflammation. Hence, the distinct phenotype of IMac-derived LPM appear to comprise: (1) predetermined features seemingly retained over time and not reprogrammed by niche signals (CD62L, Sema4a); (2) features that fail to reprogram due to an inability to compete with $RMac^{Z10}$ for environmental cues but that are reprogrammed with time (MHCII, GATA6, CCR5, FRβ); and (3) features related to time-of-residency irrespective of competition with $RMac^{Z10}$ (VSIG4, Tim4, CD209b).

**Colonizing $IMac^{Z10}$ are functionally akin to monocyte-derived RMac.** To determine whether colonizing inflammatory macrophages differ functionally and behaviorally to established resident macrophages, we developed a gating strategy based on a Tim4+ Sema4a− (R1) and Tim4−Sema4a+ (R3) profile to identify the majority of $RMac^{Z10}$ and $IMac^{Z10}$, respectively (Fig. 4a, b). Using this approach, we were able to track the major long-term changes in phenotype of the resident LPM pool triggered by inflammation without need for dye-based fate-mapping (Supplementary Fig. 4b). In addition, to determine whether IMac-derived LPM are functionally similar to LPM of recent monocyte-origin recruited during homeostasis, we confirmed that the Tim4−Sema4a+ gate identified the majority of Tim4− MHCII+ LPM in naïve mice (Supplementary Fig. 4c), which we previously validated to identify newly monocyte-derived LPM[15].

Consistent with our previous observations showing that LPM of recent monocyte-origin proliferate more than established LPM during homeostasis[21], the Sema4a+Tim4− fraction of LPM from naïve mice (subsequently referred to as Mo-LPM and RM-LPM, respectively) exhibited the highest level of proliferation, as determined by Ki67 expression (Fig. 4c). Similarly, Sema4a+ Tim4− and Sema4− Tim4+ defined-populations found 8 weeks post zymosan injection (subsequently referred to as $Mo^{Z10}$-LPM and $RM^{Z10}$-LPM, respectively) exhibited the same divergent pattern in proliferative activity (Fig. 4c). Furthermore, re-analysis of RMac obtained following transfer into macrophage-depleted recipients revealed that compared with the point of transfer (Supplementary Fig. 4d, e), the ratio of Tim4+ RM-LPM to Tim4− Mo-LPM within this population decreased over the ensuing 8 days (Supplementary Fig. 4g) due to greater expansion of the Tim4− fraction (Supplementary Fig. 4h). Indeed, while both subsets were equivalently labeled with PKH26-PCL prior to transfer (Supplementary Fig. 4f), the Tim4− fraction subsequently lost significantly more dye (Supplementary Fig. 4i), consistent with overall greater proliferation. Hence, heightened proliferative capacity appears to be a feature of monocyte-derived macrophages, irrespective of the conditions under which these cells have infiltrated the cavity.

Furthermore, while both $Mo^{Z10}$-LPM and $RM^{Z10}$-LPM displayed typical macrophage morphology, the cytoplasm of $RM^{Z10}$-LPM contained many more vacuoles (Fig. 4d) indicative of greater phagocytic activity. Indeed, both $Mo^{Z10}$-LPM and Mo-LPM had appreciably lower side-scatter characteristics than their RM counterparts, albeit higher than SPM (Fig. 4e). Moreover, examination of phagocytic potential in vitro using pHrodo-labeled *Escherichia coli* particles revealed that Tim4+ LPM from naïve mice and 8 weeks after inflammation were significantly more phagocytic than the Tim4−fraction (Fig. 4f). Of note, incubation at 37 °C for 1 h caused rapid acquisition of surface Sema4a by Tim4+ macrophages thereby preventing analysis of Sema4a-defined populations in this assay. Furthermore, re-analysis of our short-term transfer experiments revealed that only Tim4+-recipient LPM acquired PKH26-PCL dye from donor RMac irrespective of whether recipients were naïve or zymosan-injected (Supplementary Fig. 4j, k), suggesting uptake of

dying donor cells is restricted to Tim4+ cells. These data are also consistent with cell death rather than emigration from the cavity as the predominant mechanism by which excess macrophages are cleared from the cavity[33]. Lastly, to test responsiveness to challenge, RM-LPM and Mo-LPM from naïve mice and $RM^{Z10}$-LPM and $Mo^{Z10}$-LPM obtained 8 weeks post zymosan injection were purified and exposed in vitro to lipopolysaccharide (LPS) and cytokine and chemokine production assessed by multiplex assay. The overall response of $Mo^{Z10}$-LPM and Mo-LPM compared to their RM counterparts was remarkably similar; both produced higher levels of IL-10 and somewhat more IL-1β and GM-CSF and less CXCL10 and TNF (Fig. 4g, h), suggesting these are common features of monocyte-derived LPM. Furthermore, direct comparison confirmed that despite some subtle differences, Mo-LPM and $Mo^{Z10}$-LPM produced largely similar quantities of cytokines and chemokines, as did $RM^{Z10}$-LPM compared with $RM^{Z10}$-LPM (Supplementary Fig. 4l, m). Hence, together with our gene expression profiling, these data suggest that recency-of-monocyte origin more strongly influences the behavior of LPM than prior experience of inflammation and that persistence of inflammatory macrophages leads to the expansion of a normally minor subset of IL-10 producing monocyte-derived LPM present under homeostatic conditions. Finally, we found that purified $Mo^{Z10}$-LPM transferred into naïve-recipient mice produced less TNF than transferred $RM^{Z10}$-LPM upon subsequent injection of LPS (Fig. 4i, j), confirming these cells also respond differently to challenge in vivo.

**Fate of inflammatory macrophages is dependent on the severity of inflammation.** In the mild model of sterile peritonitis studied so far, the initial macrophage "disappearance reaction" and inflammatory response that occurs is relatively limited and transient[38]. In contrast, injection of a 100-fold higher dose of zymosan (1000 μg/mouse) induced an almost complete and protracted disappearance of F4/80Hi Tim4+ LPM concurrent with a greater and more protracted influx of monocytes and neutrophils[44] and overall increase in size of the CD11b+ macrophage/monocyte compartment (Supplementary Fig. 5a). Notably, the CD11b+ population remained exclusively F4/80Lo MHCIIHi for at least 11 days although Tim4+ cells had begun to re-emerge within this time (Supplementary Fig. 5b). Importantly, analysis in tissue-protected BM chimeric mice confirmed that the entire peritoneal macrophage pool, including Tim4+ cells, had been replaced from the BM 3 weeks after high-dose zymosan (Fig. 5a and Supplementary Fig. 5c). Thus, severe sterile peritoneal inflammation is a physiological setting leading to the complete ablation and replacement of resident LPM.

Hence, to understand the fate of inflammatory macrophages after severe inflammation we purified dye-negative inflammatory macrophages 3 days after injection of high- or low-dose zymosan ($IMac^{Z1000}$; Supplementary Fig. 5d) and transferred them into their native inflammatory environments. Markedly fewer donor $IMac^{Z1000}$ persisted at 8 weeks in recipients of high-dose zymosan compared with those receiving low-dose zymosan (Fig. 5b), consistent with the greater contraction in size of the peritoneal macrophage compartment (Supplementary Fig. 5a) and the reported death of the majority of inflammation-elicited macrophages that follows resolution of severe peritoneal inflammation[33,45]. However, those surviving $IMac^{Z1000}$ in the high-dose environment adopted a F4/80HiGATA6+ profile by 8 weeks following severe inflammation and almost none subsequently expressed CCR5 or FRβ (Fig. 5c–e). These results are consistent with a more mature phenotype, again reflecting more rapid differentiation in the absence of competition from enduring resident macrophages. Nevertheless, a shared deficiency

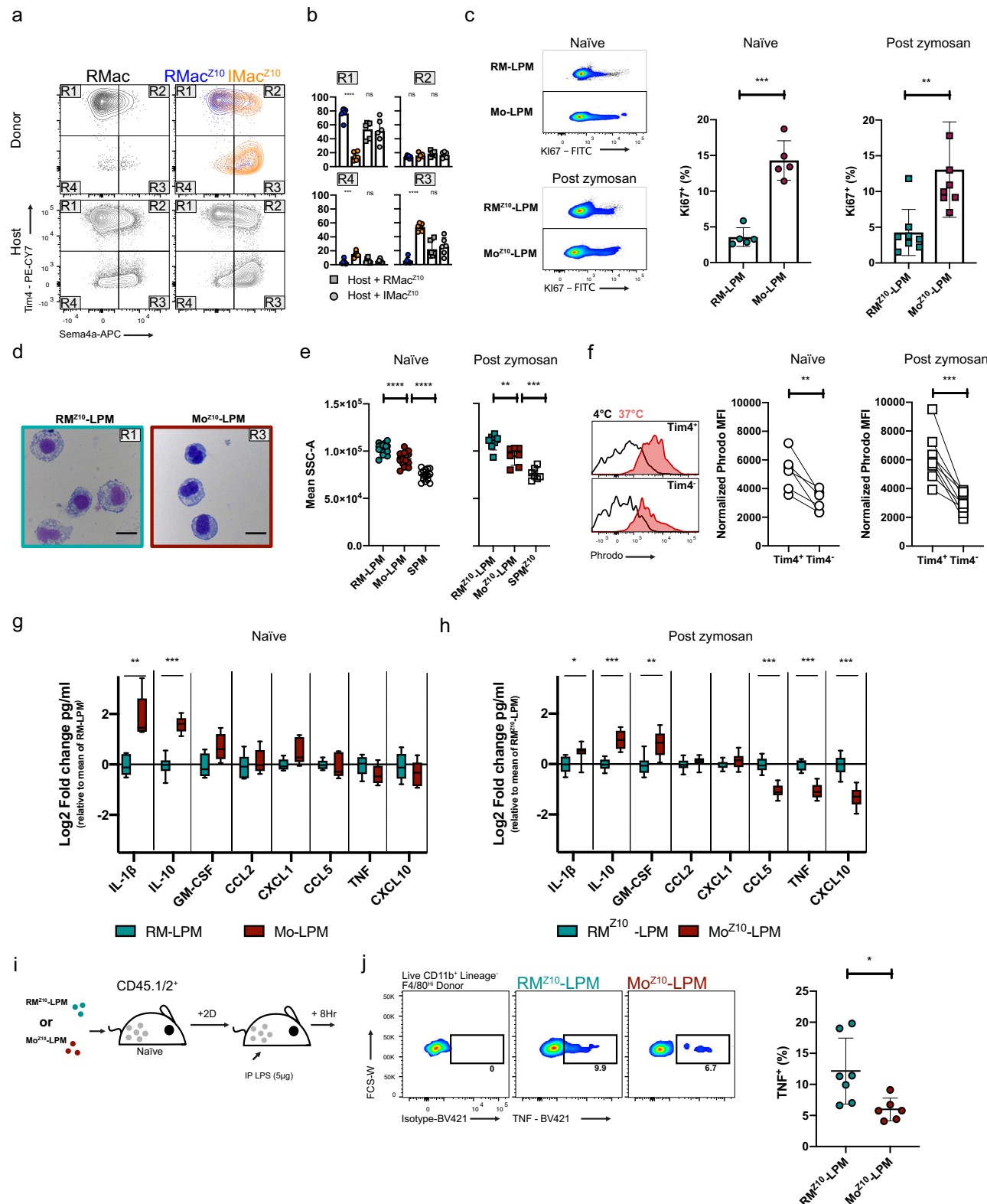

of IMac-derived cells in both high and low-dose environments was the failure to produce CXCL13, a GATA6[22,23] and RXRAB[26] independent feature of LPMs. Thus, these data suggest impaired CXCL13 expression by IMac arises from long-term alterations in the LPM niche that occurs irrespective of inflammation severity and retinoic acid production. IMac[Z1000] and IMac[Z10]-derived cells also largely shared the propensity to express the intrinsic

marker of monocyte-derived LPM Sema4a (Fig. 5e), and to lack expression of the environment-independent but time-dependent marker VSIG4 (Fig. 5e). Surprisingly, IMac[Z1000] retained high levels of MHCII, and largely expressed the otherwise time-dependent marker Tim4 (Fig. 5e). Furthermore, donor-derived macrophages following severe inflammation almost perfectly resembled the phenotype of host cells, consistent with their likely

**Fig. 4 Monocyte-derived LPM are functionally distinct from embryonically seeded LPM. a** Tim4 and Sema4a expression by RMac, RMac$^{Z10}$, and IMac$^{Z10}$ 8 weeks post transfer and host cells. **b** Proportion of RMac$^{Z10}$ (blue, $n = 6$), IMac$^{Z10}$ (orange, $n = 6$), and host (gray) macrophages with Sema4a$^{Lo}$Tim4$^+$(R1), Sema4a$^{Hi}$Tim4$^+$(R2), Sema4a$^{Hi}$Tim4$^-$ (R3), or Sema4a$^{Lo}$Tim4$^-$ (R4) phenotype. ***$p < 0.001$, ****$p < 0.0001$, one-way ANOVA and Tukey's multiple comparisons test. **c** Ki67 expression on naïve RM-LPM and Mo-LPM ($n = 5$) or 8-week post zymosan RM$^{Z10}$-LPM and Mo$^{Z10}$-LPM ($n = 8$). Naïve; $p = 0.0001$ (***), Post zymosan; $p = 0.008$(**), paired student's $t$-test. **d** Morphological appearance of RM$^{Z10}$-LPM and Mo$^{Z10}$-LPM 8 weeks post zymosan. Single experiment, scale bar 20 μm. **e** Mean side scatter of naïve RM-LPM, Mo-LPM, SPM ($n = 5$) or 8 weeks post zymosan RM$^{Z10}$-LPM, Mo$^{Z10}$-LPM, and SPM$^{Z10}$ ($n = 8$). Naïve; both $p < 0.0001$(****), post zymosan: $p = 0.0062$(**), $p = 0.0008$(***), one-way ANOVA with Tukey's multiple comparisons test. **f** Normalized Phrodo *E.coli* MFI (MFI 37 °C minus MFI 4 °C) on naïve ($n = 6$) or 8 weeks post zymosan ($n = 9$) Tim4$^+$ and Tim4$^-$ macrophages. Naïve; $p = 0.0048$(**), Post zymosan; $p = 0.0002$(***), paired student's $t$-test. **g** Analytes secreted by RM-LPM ($n = 6$, teal) or Mo-LPM ($n = 5$, red) sourced from naïve animals following 14 h LPS (1 ng/ml) treatment. Results shown as log2 fold change in mean pg/ml over mean RM-LPM. Box extends from the 25th to the 75th percentile, middle line denotes median. Whiskers denote minima and maxima. **$p < 0.001$, ***$p < 0.0001$, repeated student's $t$-test with Holm-Sidak correction. **h** Analytes secreted by RM$^{Z10}$-LPM ($n = 8$, teal) or Mo$^{Z10}$-LPM ($n = 8$, red) sourced 8 weeks post zymosan following 14 h LPS (1 ng/ml) treatment. Results shown as log2 fold change in mean pg/ml over mean RM$^{Z10}$-LPM. Box extends from the 25th to the 75th percentile, middle line denotes median. Whiskers denote minima and maxima. *$p < 0.05$ **$p < 0.001$, ***$p < 0.0001$, repeated student's $t$-test with Holm-Sidak correction. **i** Experimental scheme for purification of RM$^{Z10}$-LPM and Mo$^{Z10}$-LPM from donor mice treated 8 weeks prior with 10 μg zymosan and transfer into naïve-recipient mice followed by injection of 5 μg LPS IP. **j** Proportion of donor CD45.2$^+$ F4/80$^{Hi}$ RM$^{Z10}$-LPM ($n = 7$) and Mo$^{Z10}$-LPM ($n = 6$) 8 h post injection of LPS that express TNF. $p = 0.0206$(*), student's $t$-test. Data are presented as mean ± standard deviation with each symbol representing an individual animal. Naïve animals were age matched to zymosan-treated (15–18-week) animals. For (**c**, **g**), animals were 10–12-week at time of analysis. Data pooled from two independent experiments.

uniform origin from inflammatory macrophages (Supplementary Fig. 5e). In contrast, in the lower dose environment host macrophages neither aligned with RMac$^{Z10}$ nor IMac$^{Z10}$ but corresponded to a mixed population of these cells, (Fig. 5f and Supplementary Fig. 5e) re-emphasizing that phenotype is ontogeny-restricted in this environment. Consequently, the LPM compartment on the whole 8 weeks after high-dose zymosan differed markedly to that after low-dose zymosan for each marker assessed (Supplementary Fig. 5e).

As both F4/80 and MHCII expression by IMac$^{Z10}$ were rapidly responsive to niche signals and competition with LPM after low-dose zymosan (Fig. 1i, l), we postulated that the initially prolonged absence of F4/80$^{hi}$ LPM, rapid acquisition of Tim4 expression, and persistent expression of MHCII by recruited cells (Supplementary Fig. 5b) following severe inflammation arose from an altered cavity environment. To test this, we adoptively transferred $4 \times 10^5$ F4/80$^{Hi}$, largely MHCII$^{Lo}$, resident macrophages from naïve mice (RMac) into recipient mice 3 days after injection of high-dose zymosan (Fig. 5g). Eight days later transferred cells had almost exclusively upregulated MHCII expression and markedly downregulated expression of F4/80 (Fig. 5h, i). Moreover, transfer of RMac suppressed the rapid acquisition of Tim4 by host cells, as indicated by a specific decrease in number of Tim4$^+$ host macrophages (Fig. 5i). Hence, novel environmental cues following severe inflammation directly drive expression of MHCII, and in conjunction with the absence of embryonically seeded Tim4$^+$ resident macrophages allow rapid acquisition of Tim4 by monocyte-derived cells. Furthermore, there appears to be a phase of at least 11 days during severe peritonitis where the cavity does not support expression of F4/80 that, given the dependence of F4/80 expression by LPM on GATA6 and retinoic acid[5,22,23] (Fig. 2a, b and Supplementary Fig. 2f), suggests severe peritoneal inflammation leads to a protracted but ultimately transient loss in retinoic acid availability.

**Inflammation leads to long-term disruption of B1 cell homeostasis.** Although the failure to produce CXCL13 was a common feature of IMac-derived LPM (Fig. 5e), treatment with high-dose zymosan led to a striking reduction of CXCL13$^+$ peritoneal macrophages (Fig. 5j) arising from the comprehensive loss of the incumbent CXCL13-expressing resident cells. Given the non-redundant role of CXCL13 in maintenance of the peritoneal B1-cell pool[46], we investigated whether peritoneal inflammation led to long-term disruption of B1 cells. Temporal analysis revealed

that while the number of CD11b$^+$ B1 cells[15] gradually increased with age under homeostatic conditions (Fig. 5k and Supplementary Fig. 6a), the degree of accumulation was slightly reduced following mild inflammation and completely abrogated following severe inflammation, yet neither led to absolute loss of B1 cells over baseline levels (Fig. 5l and Supplementary Fig. 6b). Direct comparison of numbers of B1 cells in recipient mice from adoptive transfer experiments confirmed that severe inflammation led to substantially fewer CD11b$^+$ peritoneal B1 cells within the cavity than following mild inflammation (Supplementary Fig. 6c). Furthermore, severe inflammation led to increased levels of serum IgM against phosphorylcholine, the predominant target of natural antibodies produced by peritoneal B1 cells[46,47] and to the appearance of anti-phosphorylcholine IgG (Fig. 5m). Hence, sterile peritoneal inflammation leads to a state of altered homeostasis characterized by a failure to increase numbers of peritoneal CD11b$^+$ B1 cells over time but which is associated with increased levels and class-switching of circulating natural antibody. Furthermore, unlike mild inflammation, severe inflammation also led to a reduction in number of naïve CD62L$^+$ cavity T cells at 8 weeks (Supplementary Fig. 6d, e) and a low but detectable number of neutrophils (Supplementary Fig. 6d, f) despite the apparent clearance of zymosan from the peritoneal fluid (Supplementary Fig. 6g, h), suggestive of a state of persistent ongoing low-grade peritoneal inflammation.

## Discussion

Transient peritoneal inflammation has lasting consequences for the incidence and severity of future disease[40,48] but the mechanisms underlying this effect remain largely unresolved. Here, we demonstrate that inflammatory peritoneal macrophages recruited following sterile inflammation persist long-term but in an aberrant state of activation largely due to an inability to compete with incumbent macrophages for niche signals and inflammation-driven alterations in the peritoneal environment. In so doing, we reveal the existence of multiple overlapping biochemical niches that control programming of resident peritoneal macrophages and which are distinct from that controlling cell survival.

Like Liu et al.[37], our study suggests that the degree of replacement of LPM from the bone marrow following inflammation depends on the magnitude of initial macrophage disappearance. Furthermore, the increased survival of inflammatory macrophages following transfer into macrophage-depleted recipients provides definitive evidence that incumbent LPM impair survival

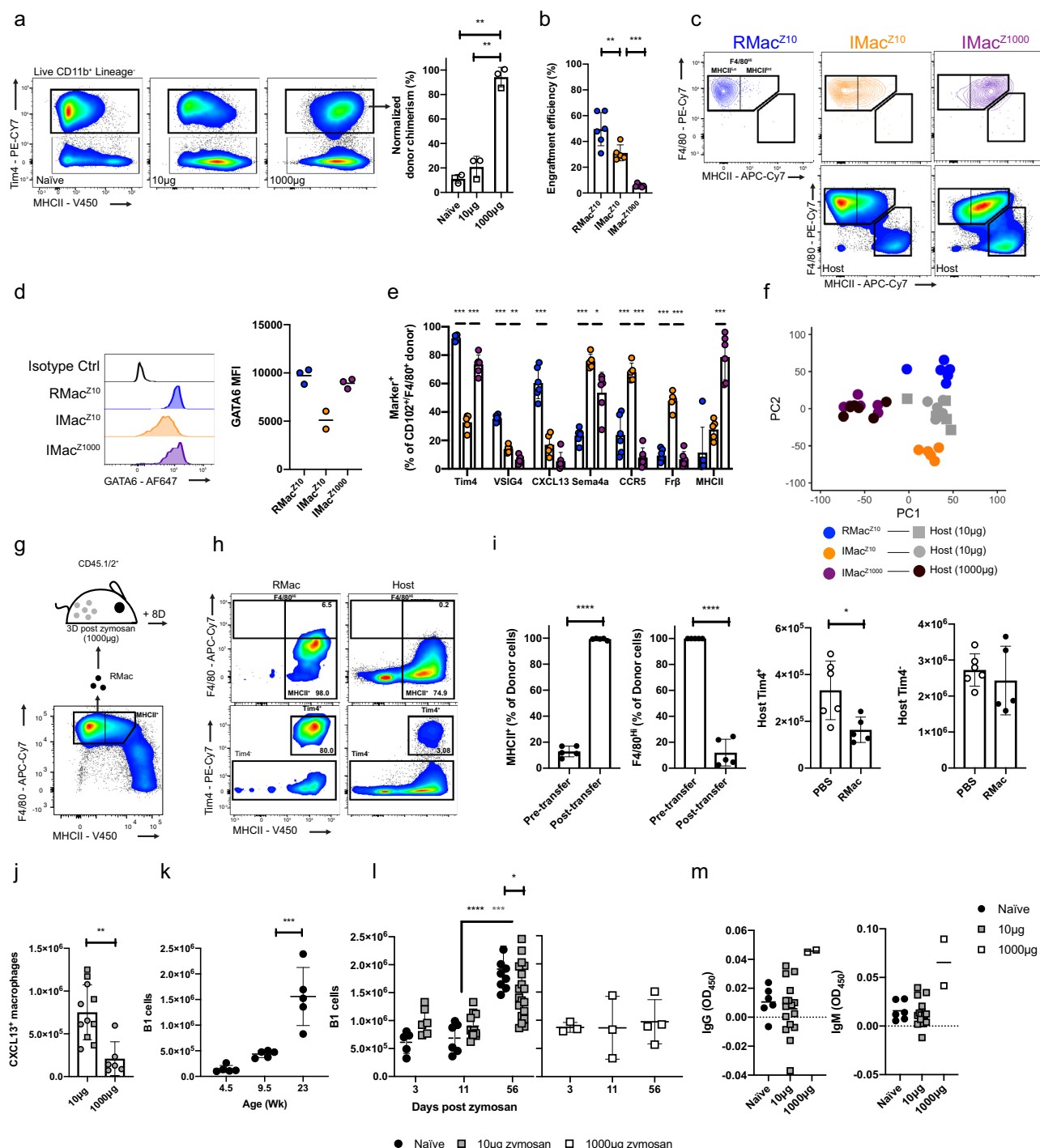

of recruited cells. Hence, even without a defined physical niche, monocyte contribution to resident macrophages within fluidic environments appears subject to the same parameters of niche access and availability proposed by Guilliams and Scott[16].

Likewise, our data support a model whereby competition for access to a biochemical niche plays a critical role in determining the long-term transcriptional identity of inflammation-elicited macrophages. Specifically, inflammation-elicited macrophages that survived following mild inflammation exhibited striking long-term differences to incumbent resident macrophages including high MHCII and low GATA6 expression, yet more rapidly adopted a GATA6$^{Hi}$ MHCII$^{Lo}$ resident-like phenotype and transcriptome following transfer into naïve macrophage-depleted mice. The

failure of inflammation-elicited macrophages to down-regulate MHCII expression following transfer into intact naïve mice confirms this feature arises from competition with incumbent resident macrophages for signals that regulate MHCII expression. Consistent with this, inflammation-elicited macrophages rapidly downregulated MHCII in vitro in response to RA-independent omental factors. Similarly, the GATA6$^{Lo}$ phenotype of inflammation-elicited macrophages following mild inflammation likely arises due to competition with enduring resident macrophages for retinoic acid, since more rapid acquisition of a GATA6$^{Hi}$ phenotype occurred following severe inflammation concurrent with the ablation of resident macrophages. Critically, inflammation-elicited macrophages gradually adopted features

**Fig. 5 Ontogeny does not control monocyte phenotype after severe peritonitis and leads to impaired B1 cell expansion. a** Normalized chimerism of Tim4[+] macrophages in tissue-protected BM chimeric mice left naïve or treated with 10 μg or 1000 μg zymosan 17 days prior ($n = 3$/group). **$p < 0.01$, one-way ANOVA and Tukey's multiple comparisons test. **b** Engraftment efficiency of RMac[Z10] ($n = 6$), IMac[Z10] ($n = 5$), and IMac[Z1000] ($n = 6$), after transfer into the mirroring recipients. $p = 0.005$(**), $p = 0.00038$(***), one-way ANOVA and Tukey's multiple comparisons test. **c** Expression of F4/80 and MHCII on indicated populations 8 weeks post after transfer. **d** GATA6 MFI of donor RMac[Z10], IMac[Z10] and IMac[Z10] after transfer into native recipients ($n = 3, 2, 3$). **e** Marker expression by CD102[+] or F4/80[+] donor RMac[Z10] (blue), IMac[Z10] (orange), and IMac[Z1000] (purple) 8 weeks after transfer into mirroring recipients ($n = 6, 5, 6$). *$p < 0.05$, **$p < 0.01$, ***$p < 0.001$, one-way ANOVA and Dunnet's multiple comparisons test for each marker, followed by Bonferroni adjustment. **f** Principal component analysis based on all markers assessed in **e**. **g** Scheme for transfer of F4/80[Hi] MHCII[Lo] naïve resident macrophages (RMac) into mice injected 3 days prior with 1000 μg zymosan. **h** F4/80, Tim4, and MHCII expression on RMac and host myeloid cells 8 days post transfer. **i** Proportion of RMac that express MHCII and high levels of F4/80 before transfer and 8 days post transfer ($n = 5$; left). Number of host Tim4[+] or Tim4[-] macrophages 8 days post transfer ($n = 5$). In order: $p < 0.0001$ (****), $p < 0.0001$ (****), $p = 0.021$ (*), student's $t$-test. **j** Number of CXCL13[+] host macrophages 8 weeks after 10 μg ($n = 11$) or 1000 μg zymosan ($n = 6$) treatment. $p = 0.0020$ (**), student's $t$-test. **k** Number of peritoneal B1 cells in naïve female mice at indicated age in weeks ($n = 5$/timepoint). $p = 0.00047$(***), one-way ANOVA and Tukey's multiple comparisons test. **l** Number of peritoneal CD11b[+] B1 cells at indicated timepoints in naïve mice (black; $n = 5, 6, 10$) or mice treated with 10 μg (gray; $n = 6, 11, 20$) or 1000 μg zymosan (white; $n = 3, 3, 4$). *$p < 0.05$, ***$p < 0.001$, ****$p < 0.0001$, two-way ANOVA and post hoc Tukey test. **m** Serum anti-phosphorylcholine IgM and IgG in naïve mice ($n = 6$), or mice treated with 10 μg ($n = 16$) or 1000 μg ($n = 2$) zymosan 8 weeks prior. Data are presented as mean ± standard deviation with each symbol representing an individual animal. Data were pooled from at least two independent experiments except high-dose treatment (**l**, **m**), which is a single experiment. Host cells represented by squares or circles are recipients of RMac[Z10] or IMac[Z10], respectively.

seemingly regulated by competition, suggesting that with time these cells receive sufficient cues to acquire a mature resident phenotype.

Several features of inflammation-elicited macrophages appeared regulated by changes in the cavity micro-environment post inflammation. For example, severe inflammation led to sustained expression of MHCII by inflammation-elicited macrophages despite natural ablation of competing resident cells. Furthermore, the severely inflamed environment triggered MHCII expression by adoptively transferred resident LPM that would otherwise remain largely MHCII[-] in a non-inflamed or mild inflammation setting. Hence, it seems likely severe inflammation triggers release of novel signals stimulatory for MHCII.

A small number of genes remained differentially expressed in inflammation-elicited macrophages following transfer into macrophage-depleted mice, supporting the notion that developmental origin influences macrophage identity[49]. These included features reprogrammed over time (e.g., TIM4, VSIG4, CD209b), and permanent "legacy" features of monocyte-derived cells (e.g., Sema4a, CD62L). The processes regulating these traits remains unclear[49]. However, the rapid acquisition of Tim4 expression by inflammation-elicited LPM following severe inflammation and the inhibition of this by transfer of Tim4[+] resident macrophages provides proof-of-principle that seemingly origin-related time-dependent features of resident macrophages can be rapidly reprogrammed by appropriate niche signals.

While the panel of genes assessed here was relatively limited, our findings that gene expression is largely dictated by competition with resident cells or the post-inflammatory environment are likely to hold true on the transcriptome as a whole. For example, the overlap between environment-dependent genes and GATA6 regulated genes[5,22,23] suggests lower GATA6 expression by inflammation-elicited macrophages controls a significant proportion of their unique transcriptional profile. Similarly, the retarded expression of GATA6 by monocyte-derived LPM macrophages recruited during homeostasis also likely underlies a significant degree of the distinct transcriptional clustering of these cells. Critically, these data reveal that GATA6 expression does not act binarily but rather the level of expression has a critical role in determining LPM identity, as predicted for transcription factors with many competing target sites[50].

One of our most intriguing findings was the difference in proliferative activity of resident and monocyte-derived LPM, with only the latter exhibiting the capacity to overtly expand following transfer into macrophage-depleted mice. Notably, GATA6

directly regulates proliferation of LPM[23], suggesting this disparity may relate to differences in GATA6 expression. However, as treatment with exogenous CSF1 or IL-4 stimulates recently recruited and incumbent resident macrophages to proliferate to equivalently high degrees[21,37,51], the poor expansion of incumbent resident macrophages within the macrophage-deplete environment cannot be due to an intrinsic inability to proliferate. Rather, GATA6-deficient macrophages have an enhanced ability to interact with mesothelium[5], suggesting monocyte-derived LPM may have a greater ability to utilize the membrane-bound CSF1 produced by these cells[52].

Despite differentiating under inflamed conditions, persistent inflammation-elicited LPM bore striking similarities to monocyte-derived LPM present under non-inflamed conditions. As well as gene expression and proliferative activity, monocyte-derived LPM exhibited a largely comparable response to LPS irrespective of condition of differentiation, most notably characterized by increased production of IL-10 compared to embryonically seeded LPM. Other than IL-1β, the profile of cytokine production by monocyte-derived LPM was the opposite reported for GATA6-deficient LPM[53], suggesting other factors control their differential response to LPS. Hence, mild peritonitis does not lead to the existence of a unique subset of LPM but rather the expansion of a subset present in homeostasis. As we have previously shown that the abundance of Tim4[-] monocyte-derived LPM gradually increases with age[21], it would appear that mild inflammation accelerates a process normally associated with aging.

Our data also predict that inflammation-elicited LPM fail to express *Cxcl13* due to inflammation-induced loss of requisite niche signals. Notably, *Cxcl13* expression by LPM is sustained in vitro without addition of niche factors[43], potentially explaining why CXCL13 expression remains intact in incumbent LPM following mild inflammation. Hence, unlike the reversible program of gene expression controlled by GATA6 that is lost in the absence of RA[43], niche signals required to induce expression of CXCL13 in newly recruited macrophages may not be needed to maintain expression in resident cells.

*Cxcl13*-deficient mice are profoundly deficient in peritoneal B1 cells[46] yet CXCL13 is not required for retention of B1 cells in the cavity[54]. We found the extent of inflammation and consequently the ratio of CXCL13-producing resident macrophages to monocyte-derived CXCL13[-] LPM in the cavity correlates with a failure to accumulate peritoneal B1 cell with age. As no other peritoneal lavage cells produce CXCL13[15,46], our data suggest CXCL13 production by peritoneal macrophages is required for

continued recruitment of B1 cells from the circulation. Notably, replacement of peritoneal LPM by monocytes and concurrent failure to expand peritoneal B1 cells also occurs following abdominal surgery[15]. Hence, long-term dysregulation of B1 cells is likely a general feature of peritoneal inflammation. Sterile peritoneal inflammation also led to elevated circulating natural antibody. Whereas splenic and bone marrow B1 cells spontaneously secrete high levels of natural IgM, those in the cavities do not[55]. Furthermore, levels of serum anti-PC IgM gradually drop with age[56]. Hence, we speculate that the failure to accumulate B1 cells in the peritoneal cavity may allow their re-entry into tissues permissive for antibody secretion such as fat associated lymphoid clusters[57].

In addition to perturbed homeostasis of B1-cells, high-dose zymosan treatment led to a significant long-term elevation in number of peritoneal neutrophils and decrease in number of naïve CD62L[+] peritoneal T cells, suggesting persistent low-grade peritoneal inflammation. Whether this represents a permanent alteration in tissue homeostasis and if so whether this arises from innate or adaptive immune memory[58] or dysregulation of normal feedback loops between tissue T cells and macrophages[59] remains unclear, but notably, loss of peritoneal CD62L[+] T-cells has been implicated in low-grade peritoneal inflammation and induction of MHCII-expression on peritoneal macrophages[27]. Furthermore, we cannot at present exclude the possibility that these alterations in immune cell homeostasis are driven by the continued presence of an undetected reservoir of zymosan.

Our study highlights how varying degrees of inflammation alter the peritoneal macrophage compartment long-term and consequently reshape peritoneal homeostasis, and implicates competition for niche signals, time-of-residency and alterations in niche as principal determinants of these phenomena. These findings have broad importance for our understanding of plasticity within the mononuclear phagocyte compartment. Furthermore, understanding the consequences of inflammation in the serous cavities has major implications for pathologies in which serous cavity macrophages play key roles, including endometriosis[60], adhesions[61], and repair and scarring of visceral organs[20,62] and the myriad of diseases influenced by natural antibody[63].

## Methods

**Mice and reagents**. C57BL/6JCrl and congenic CD45.1[+]CD45.2[+] mice (B6.SJL-PtprcaPep3b/BoyJ × C57BL/6JCrl) were bred and maintained in specific pathogen-free facilities at the University of Edinburgh, UK. In some experiments, C57BL/6JCrl mice were purchased from Charles River, UK. Mice were sex matched in all experiments and used at 6–10 weeks of age at the start of the experiment. All animal experiments were performed under license by the UK Home Office and received ethical approval from the University of Edinburgh Animal Welfare and Ethical Review Body. All animal experiments were performed in accordance with the ethical regulations for animal testing and research as set out by the UK Animals (Scientific Procedures) Act of 1986. Details of reagents and antibodies used can be found in Supplementary Tables 1 and 2.

**Sterile peritoneal inflammation**. To elicit sterile peritoneal inflammation, mice were injected with 10 or 1000 μg of zymosan A (Sigma-Aldrich) suspended in 200 μl Dulbecco's PBS (Invitrogen), dPBS or left naïve as indicated. In some experiments, mice were injected intraperitoneally with 250 μl of 700 nm PKH26-PCL in suspended in Diluent B (Sigma) 24 h prior to zymosan. In some experiments, mice were injected intraperitoneally with 0.0625 mg Clodronate liposomes (Liposoma) suspended in 250 μl dPBS (Gibco). To elicit LPS-induced inflammation mice were injected intraperitoneally with 5 μg of LPS (O111:B4, Sigma-Aldrich) suspended in 200 μl dPBS.

**All trans retinoic acid administration**. Stock ATRA was prepared in DMSO (40 mg/ml; both Sigma-Aldrich). On the day of injection a working solution was prepared by diluting ATRA stock in sterile corn oil (Sigma-Aldrich) to a concentration of 8.33 μg/μl. Mice received IP injection with 30 μl of stock solution IP at indicated timepoints, equating to 250 μg per injection.

**Tissue-protected BM chimeric mice**. Eight week-old female C57BL/6J CD45.1[+]CD45.2[+] or CD45.2[+] C57BL/6J mice were exposed to a single dose of 9.5 Gy γ-irradiation under anesthetic, with all but the hind legs of the animals protected by a 2-inch lead shield. Animals were subsequently given 2–5 × 10[6] BM cells from female congenic CD45.2[+] C57BL/6J or C57BL/6J CD45.1[+]CD45.2[+] animals, respectively, by i.v. injection and then left for 8 weeks, or in one experiment for 26 weeks due to the COVID-19-pandemic, before injection of zymosan.

**Cell isolation and flow cytometry**. Mice were sacrificed by exposure to rising levels of CO2. The peritoneal cavity was lavaged with a total of 9 ml wash solution (dPBS containing 2 mM EDTA,1 mM HEPES) or 9 ml culture solution (RPMI containing 1 mM HEPES) if cells were used for subsequent cell culture experiments. In some experiments, blood was then taken from the inferior vena cava or the carotid artery and serum isolated using Microtainer SST tubes (BD). Serum was frozen at −80 °C before analysis by ELISA. For chimeric mice, blood samples were also taken on the day of necropsy by cardiac puncture or by cutting the carotid artery. Equal cell numbers where incubated at room temperature for 10 min with zombie aqua viability dye (BioLegend), followed by 10-min incubation on ice with blocking buffer containing 10% mouse serum with 0.25 μg/ml anti CD16/CD32 (BioLegend). Cells where incubated with indicated antibodies (Supplementary Table 2) on ice for 30 min. Cells where washed with FACS buffer (2 mM EDTA/ 0.5%BSA in PBS) and if applicable stained with streptavidin conjugated or secondary antibody. For intracellular staining, cells where fixed/ permeabilized using the Foxp3 staining buffer (eBioscience) according to the manufacturers protocol. For intracellular cytokine staining 1E6 peritoneal cells were incubated for 4.5 h at 37 °C in 200 μl sterile RPMI containing Brefeldin A and Monensin (both 1:1000) in cell repellent 96-well plates (Greiner Bio-One) after which cells were washed once and stained on ice for extracellular and intracellular as described with one additional Fc blocking step (10 min on ice) after fixation. Samples were acquired using FACS LSRFortessa (BD) and analyzed using Flowjo (Version 10.4.1,Treestar). For analysis doublets (on the basis of Forward scatter area vs height) and dead cells (ZombieAqua positive) were excluded. For cell sorting, cells were stained using the same protocol scaled accordingly DAPI was used as viability dye to ensure real time viability detection. For adoptive transfer studies and cell culture studies, cells were sorted using a FACSFusion of FACSAria sorting system with a 100 μm sort nozzle. Cells sorted for RNA extraction were sorted using a 70 μm nozzle.

**Adoptive transfer of macrophages**. Cells were kept on ice at all time and all steps were carried out in a laminar flow hood using sterile reagents. Cells were collected and stained as described above and were sorted using flow cytometry into the indicated populations. Post sort cells were pelleted (300 × g, 5 min at 4 °C), resuspended in 200 μl of dPBS and counted by Casy Counter (Scharfe). For short-term and long-term studies 1 × 10[5] and 1 × 10[5] cells of the indicated populations were transferred respectively. For transfer of RMac into high-dose zymosan-treated recipients, 4 × 10[5] cells were used. To study responsiveness to LPS in vivo, 2.5 × 10[5] cells of each of the indicated populations was transferred. If required, purified populations from multiple donor mice were pooled. For each experiment, purified cells were suspended in 200 μl of dPBS and injected intraperitoneally into recipients. Survival of donor cells throughout the manuscript is presented as engraftment efficiency, which is the No. of donor cells transferred/No. of donor cells retrieved*100.

**NanoString assay**. For each of the indicated donor populations 5000 cells of interest were sorted into 2 μl of RLT (QIAGEN). Immediately after cells were centrifuged at maximum speed 15 s, vortexed for 10 s and again centrifuged at maximum speed 15 s. Cells where stored at −80 °C until analysis using the nCounter Myeloid innate immunity panel (NanoString) according to the manufacturer's instructions. Data was analyzed using the nSolver advanced analysis package. Differential gene expression was determined by pairwise comparison of IMac[Z10] and RMac[Z10] to RMac. Benjamini Hochberg adjusted p-value < 0.05 were considered differentially expressed. Figures were generated using R packages GGplot2, Pheatmap, EnhancedVolcano and GOplot.

**Gene expression ranking analysis**. Following NanoString analysis, the normalized messenger RNA gene-counts for each donor population following transfer into mirroring or clodronate-depleted recipients were collected. For each sample all detected genes were ordered on the basis of their expression levels. Each gene was assigned a percentile rank based on the rank in the ordered gene list. Then a delta gene rank percentile was calculated for each donor population (Percentile rank in native environment—Percentile rank in depleted environment) and plotted. This analysis was adapted from Galatro et al.[64].

**Gene-set enrichment analysis**. Gene set of GATA6-regulated genes was obtained by analyzing GSE56711, GSE37448, and GSE47049 using the GEO2R. Genes were considered GATA6 regulated if they were differentially expressed (p-adj < 0.05) in 2 out of 3 published GATA6[KO] datasets[5,22,23]. The gene list was split into GATA6[KO] up and downregulated gene sets. GSEA was carried out using GSEA desktop 4.1 (Broad Institute). RNA levels of genes in RMac[Z10] and IMac[Z10] were analyzed using default settings and 10.000 geneset permutations. A gene set of RXRAB-regulated genes was obtained by analyzing GSE129095 using the GEO2R

tool. Genes were GATA6-independently, RXRAB-regulated if they were differentially expressed (*p*-adj < 0.05) in GSE129095 but were not regulated by GATA6. The resulting gene list was split into RXRAB[KO] up and downregulated gene sets for GSEA analysis.

**Omentum factors production and treatment**. Omentum factors were generated by culturing the omentum from naïve mice in 1 ml of macrophage serum-free media (GIBCO) for 5 days as described[5] after which medium was collected, centrifuged at $300 \times g$ and the supernatant collected and diluted in 1:2 in media. Peritoneal cells were collected 11 days post low-dose zymosan were collected as described under sterile conditions and $5 \times 10^5$ plated and incubated for 2 h at 37 °C in cell culture medium (RPMI, 10%FCS, 1% L-Glutamine and 1% penicillin/streptomycin supplemented with 20 ng/ml CSF1) after which medium was aspirated and cells were incubated in 250 µl cell culture medium with 250 µl Omentum factors or macrophage serum-free media with ATRA (Sigma, 1 µm) or without for 24 h. Then, medium was removed and plate was incubated with 5 mM EDTA in ice-cold PBS for 10 min on ice to collect cells. Wells were repeatedly washed with ice-cold 5 mM EDTA PBS and wells were inspected using a microscope to confirm negligible adherent cells remained. Cells were quantified and prepared for flow cytometry as described.

**Cytokine production assay**. Cells were kept on ice at all time and all steps were carried out using sterile reagents in a laminar flow hood using the sort protocol described. For each population of interest $1 \times 10^3$ cells per condition were sorted into 75 µl sort medium (folic acid-deficient RPMI containing 20% FCS (low endotoxin), 2% L-Glutamine and 2% penicillin/streptomycin). Cells were centrifuged at $100 \times g$ at 4 degrees for 5 min. The total mixture was then transferred into a 96-well plate incubated at 37 °C for 2 h. Media was gently aspirated and cells were resuspended in 75 µl cell culture medium (folic acid-deficient RPMI supplemented with 1 µg/ml Folic Acid (Sigma-Aldrich), 10% FCS (low endotoxin), 1% L-glutamine and 1% penicillin/streptomycin). Where indicated cells received a final concentration of LPS of 1 ng/ml (O11:B4, Sigma-Aldrich) in cell culture medium or equivalent amount of dPBS. Cells were incubated for 14 h and supernatant was collected and analyzed for cytokine release using the Legendplex Mouse Anti-Virus or Mouse Inflammation panel according to the manufacturers protocol. Data was acquired using an Attune flow cytometer and analyzed using the Legendplex analysis software.

**Phrodo phagocytosis assay**. Cells were collected and $2 \times 10^6$ cells/sample were stained as described. Each sample was washed twice with ice-cold RPMI and was split into two tubes each and left on ice for 10 min. Then to each tube 10 µl of Phrodo *E. coli* particles was added and for each sample one tube was incubated at 37 °C and one at 4 °C for 1 h. All samples were placed on ice and washed once using 300 µl Buffer C and were then resuspended in 300 µl Buffer C. Cells were analyzed directly after finishing the protocol. Data is presented as normalized Phrodo mean fluorescence intensity (MFI 37 °C minus MFI 4 °C).

**Enzyme-linked immunosorbant assays**. Ninety-six-well flat-bottom high-binding polystyrene plates (Corning) were coated with 50 µl of 2 µg/ml phosphorylcholine conjugated to BSA (PC-BSA; 2B Scientific) diluted in PBS at 4 °C overnight. Plates were then blocked with 100 µl of blocking buffer (1% Casein in PBS; VWR) for 1.5 h at room temperature, before serum samples were added at 1:100 dilution in 50 µl blocking buffer and incubated for 2 h at room temperature. Wells without antigen were used as blank controls for each sample to measure non-specific antibody binding. Plates were then incubated with 1:5000 HRP-conjugated anti-mouse IgG (Abcam) or 1:2000 anti-mouse IgM (Southern Biotech) in blocking buffer for 1 h at room temperature before addition of 3,3′,5,5′-tetramethylbenzidine (TMB) (Seracare). After 10 min the reaction was stopped with 0.16 M sulfuric acid solution and the $OD_{450}$ value measured. Values for blank controls were then subtracted for each sample to quantify antigen-specific antibody levels. Plates were washed twice with 0.1% Tween20 (Sigma-Aldrich) in PBS between all steps except before addition of TMB, when they were washed five times.

**Statistics**. Statistics were performed using Prism 7/8 (GraphPad Software). The statistical test used in each experiment is detailed in the relevant figure legend. For all *t*-tests performed in this manuscript the two-tailed *p*-value is presented and interpreted.

**Reporting summary**. Further information on research design is available in the Nature Research Reporting Summary linked to this article.

## Data availability

Nanostring data that support the findings of this study have been deposited in the Gene Expression Omnibus under the accession code GSE165036. Previously published publicly available datasets analyzed in this study are available in Gene Expression Omnibus under the accession codes GSE56711, GSE37448, GSE47049, and GSE129095. Source data are provided with this paper. All other data that support the findings of this study are available from the corresponding author upon reasonable request. Source data are provided with this paper.

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

## Acknowledgements

Flow cytometry data were generated with support from the QMRI Flow Cytometry and Cell Sorting Facility, University of Edinburgh. NanoString was performed by the Host and Tumor Profiling Unit Services Facility, MRC Institute for Genetics and Molecular Medicine, University of Edinburgh. We thank Dr. Mohini Gray for kind provision of PC-BSA and Prof Judith Allen for comments on the manuscript. This work was supported by the Wellcome Trust (108906/Z/15/Z) with additional support from the Medical Research Council UK (MR/L008076/1 to S.J.J.).

## Author contributions

P.A.L. designed and performed most experiments, analyzed, and interpreted the data, and wrote the manuscript. L.B.G. performed and analyzed antibody ELISA's. H.W. and G.P.-W. performed experiments. C.C.B. contributed to design of experiments, interpretation of data, and critiqued the manuscript for intellectual content. S.J.F. provided critical feedback on the study design and critiqued the manuscript for intellectual content. S.J.J. conceived, designed, and performed experiments, analyzed and interpreted the data, wrote the manuscript, and supervised the project.

## Competing interests

The authors declare no competing interests.
