## [Peer Review File · Nature Communications]

REVIEWER COMMENTS

Reviewer #1 (Remarks to the Author):

In this manuscript the authors study the replenishment of the peritoneal macrophage niche during inflammation. The authors utilize well-designed transfer experiments to conclusively determine the effect of origin, time of residence and inflammation exposure on the capacity to colonize the peritoneal macrophage niche. The concept of more rapid differentiation of monocytes in absence of competition from resident macrophages is convincingly demonstrated and will probably be widely applicable in the macrophage field.

This study is well done and solid. I congratulate the authors for this work.

I have only one minor comment:

1) Could the authors show a gating strategy for B1 B cells and a flow plot of B1 B cells in the different mice in Figure 4K and 4I.

Reviewer #2 (Remarks to the Author):

Louwe et al, have performed an sophisticated study examining the macrophages of the peritoneal cavity during inflammation, these cells appear to have defined patterns of renewal and function depending on resolving or chronic peritonitis. The authors have implemented an ingenious combination of strategies to tease out the impact of chronic and acute inflammation upon resident and infiltrating macrophages. The study is clearly written with an extensive introduction and discussion.

The authors use an astute combination of cell labelling, flow cytometry and congenic adoptive transfer studies to tease out the contribution of each macrophage population during peritonitis. The initial set of experiments conducted in (Fig1) confirms the robust/reproducible nature of this approach. For unity in Fig1a please add the control F4/80 Ly6C plot.

It remains confusing what happens when either RMACZ10 or IMACZ10 are transferred alone, can one population transform into the other overtime? Interestingly, the burst of proliferation observed in IMACZ10 when transferred into clodronate depleted mice does this proliferation by dilute the PKH26 expression Fig1h in a measurable fashion? If so, could the number of divisions be calculated? It would be helpful to move the GATA6 data from the supplementary to Fig1. The percentage of cells should be explained within in the text this would be helpful to understand the comparison from what was adopted to cells analysed overtime. Also, do the authors know where the cell that disappear go? Do they undergo apoptosis or drain to lymph nodes?

While examining if IMACZ10 persist long-term in Fig2e the black open histogram is this an Isotype control or FMO? Please mention this in the Figure legend. To determine the expression levels of GATA6 could RMAC be included for comparison. How does the reduction in GATA6 expression on the IMACZ10 and the genes it regulates impact on function? Do the authors know how these cells differ from RMAC at steady state and during a second inflammatory insult?

It is interesting that following a second insult it is LPM-of recent monocyte origin that proliferate more, measured by ki67. Had the authors considered using EdU to measure proliferation and define when the burst of proliferation occurs or pHistoneH3 as Davies et al EJI 2011 have previously performed? Also, could the intensity of the PKH26 be used to define the extent of proliferation? Are these monocyte-LPM following inflammation functioning as repair/resolution macrophages? If so it would interesting to know how the inflammatory microenvironment is unable to redirect their function of these highly plastic cells, and could be discussed.

It is no great surprise that the magnitude of inflammation impacts on the inflammatory response. However, it is unexpected how this influences the post inflammation macrophage pool long-term.

Unless does inflammation truly vanish with such a large dose of zymosan? Is it possible to confirm all the zymosan particles have been cleared? Alternatively, is the peritoneal cavity in a perpetual state of chronic inflammation? The replacement of LPM following Z1000 and their survival/development into GATA6 positive macrophages is interesting yet these cells don't express CXCL13 like IMACZ10, leading to an altered homeostasis of resident B cells. A final question, maybe for future studies could administration of CXCL13 rescue B cell homeostasis.

Typo

1.Line 272 Z missing from RMAC10

We thank the reviewers for their constructive comments on the manuscript. Based on these we have now reworked the text and the figures and provide a point-by-point response to the comments and questions below.

Reviewer 1

In this manuscript the authors study the replenishment of the peritoneal macrophage niche during inflammation. The authors utilize well-designed transfer experiments to conclusively determine the effect of origin, time of residence and inflammation exposure on the capacity to colonize the peritoneal macrophage niche. The concept of more rapid differentiation of monocytes in absence of competition from resident macrophages is convincingly demonstrated and will probably be widely applicable in the macrophage field.

This study is well done and solid. I congratulate the authors for this work.

-- We thank the reviewer for acknowledging the importance and rigour of our manuscript.

I have only one minor comment:

1) Could the authors show a gating strategy for B1 B cells and a flow plot of B1 B cells in the different mice in Figure 4K and 4L.

-- B1 cells were identified as CD11b⁺ MHCII⁺ cells within the Lineage (CD3, CD19, Ly6G, SiglecF) gate since MHCII expression is restricted to the CD19⁺ B cells within this gate and CD11b expression discriminates mature B1 cells from B2 B-cells in the peritoneal cavity (PMID: 11825566; PMID: 27582256; PMID: 18375763). The gating strategy for CD11b⁺ B1 cells and representative flow plots for mice shown in Fig4K and L are now included in Supplementary Figure 6a,b.

Reviewer 2

Louwe et al, have performed an sophisticated study examining the macrophages of the peritoneal cavity during inflammation, these cells appear to have defined patterns of renewal and function depending on resolving or chronic peritonitis. The authors have implemented an ingenious combination of strategies to tease out the impact of chronic and acute inflammation upon resident and infiltrating macrophages. The study is clearly written with an extensive introduction and discussion.

The authors use an astute combination of cell labelling, flow cytometry and congenic adoptive transfer studies to tease out the contribution of each macrophage population during peritonitis. The initial set of experiments conducted in (Fig1) confirms the robust/reproducible nature of this approach.

-- We thank the reviewer for their positive comments.

For unity in Fig1a please add the control F4/80 Ly6C plot.

-- We have now added the control F4/80 Ly6C plot to Figure 1a.

It remains confusing what happens when either RMACZ10 or IMACZ10 are transferred alone, can one population transform into the other overtime?

-- We apologise that the description of our methodology was unclear. In all adoptive transfer experiments RMac^{Z10} and IMac^{Z10} were transferred into separate recipient mice to allow subsequent fate mapping of these cells using CD45.2 expression alone. To clarify this, we have now added the word "or" to each of the experimental design graphics in Fig. 1c, 2a, and 3i to highlight that recipient mice received RMac^{Z10} 'or' IMac^{Z10}. Hence, our flow-cytometry analysis suggests that IMac^{Z10} slowly acquire features of RMac^{Z10} following transfer into the native inflamed environment and that this occurs more rapidly in the non-inflamed macrophage-depleted environment (Fig. 1 and Fig. 2). However, to determine if this is also the case at transcriptional level, we now include gene-rank analysis of the NanoString targeted transcriptional profiling of these cells (supplementary Figure 3j). This analysis demonstrates that the gene expression profile in RMac is relatively unaffected by transfer into the non-inflamed macrophage-depleted environment whereas the expression profile of IMac^{Z10} changes much more significantly. Hence, this demonstrates that the transcriptional coalescence of IMac^{Z10} and RMac^{Z10} that occurs upon transfer into the macrophage-depleted environment is largely caused by transformation of IMac^{Z10} into RMac^{Z10} rather than the transformation of RMac^{Z10} into IMac^{Z10}. We have now included this data in Supplementary Figure 3j. and added additional text at lines 313-318

Despite this, we also noted that early following transfer of RMac^{Z10} into macrophage-depleted mice, there was an apparent marginal reduction in Tim4-expressing cells and increase in MHCII-expressing cells when compared to transfer into the native environment (Supplementary Fig2c native vs depleted), suggesting RMac^{Z10} may acquire some features of IMac^{Z10} in this environment. However, while addressing the next reviewer's comment we re-assessed whether the PKH26-PCL dye-label used for short-term identification of RMac^{Z10} could also track cell proliferation history. This revealed that Tim4⁻ RMac^{Z10}, which represent recently monocyte-derived cells (PMID: 32561560, PMID: 27292029), proliferated more rapidly and accumulated to a greater degree than Tim4⁺ RMac^{Z10} following transfer into the macrophage-depleted environment (this data has now been added as SFig4d-i). Hence, while the increase in proportion of Tim4⁻ cells gives the impression that the RMac^{Z10} population can acquire features of IMac^{Z10} in the macrophage-deplete environment, we can show this actually arises from the transient out-competition of established resident macrophage within RMac^{Z10} by the Tim4⁻ monocyte-derived component. Overall then, our data indicate that in the steady-state or following resolution of inflammation, IMac can gradually acquire features of RMac but RMac do not acquire features of IMac, and hence, differentiation of peritoneal macrophages follows a linear trajectory from a transcriptional state associated with monocyte-derived cells to that of one associated with established resident macrophages.

Interestingly, the burst of proliferation observed in IMACZ10 when transferred into clodronate depleted mice does this proliferation by dilute the PKH26 expression Fig1h in a measurable fashion? If so, could the number of divisions be calculated?

-- We thank the reviewer for this great suggestion. Unfortunately, we could not track proliferation of IMac^{Z10} using dilution of the PKH26 dye since only RMac^{Z10} are labelled in our methodology (see Fig 1a). However, as mentioned, we used Tim4 expression to split the RMac^{Z10} population into its monocyte-derived (Tim4⁻) and established LPM (Tim4⁺) components (PMID: 32561560, PMID: 27292029), and found that the Tim4⁻ cells

expanded in number more and subsequently lost more of the PKH26 label following transfer into the clodronate depleted mice (SFig4d-i). Although loss of dye is a useful secondary readout of proliferation, calculating the exact number of divisions on the basis of this PKH26-PCL is imperfect as PKH26-PCL is retained in the phagolysosomes which may not be distributed equally between daughter cells following cell division (PMID: 18795151). Hence, using this approach has allowed us to verify our conclusions drawn on Ki67 expression (that the monocyte-derived fraction of RMac proliferate more rapidly than established cells). These data have now been added as Supplementary Fig4d-i.

It would be helpful to move the GATA6 data from the supplementary to Fig1.

-- As suggested, we have moved the data to Figure 1.

The percentage of cells should be explained within in the text this would be helpful to understand the comparison from what was adopted to cells analysed overtime.

-- We thank the reviewer for pointing out this ambiguity. We have now amended the text in the methods to make clear how engraftment efficiency was calculated.

Also, do the authors know where the cell that disappear go? Do they undergo apoptosis or drain to lymph nodes?

-- It is likely that isolation, FACS purification and injection results in the subsequent death of at least some of the adoptively transferred cells and consistent with this, we demonstrated in the original manuscript that low levels of donor cell-restricted PKH26 label could be found within a significant proportion of recipient macrophages (now SFig. 4j). Furthermore, while we have not directly assessed migration of donor cells to the lymph nodes, the removal of excess macrophages that occurs during the resolution phase of inflammation has been shown to largely be due to cell-death whereas migration to the draining lymph nodes plays a lesser role (PMID: 23974197). We have amended text in the result section including reference to PMID: 23974197 to make it clearer that apoptosis contributes to the initial loss of donor cells.

While examining if IMACZ10 persist long-term in Fig2e the black open histogram is this an Isotype control or FMO? Please mention this in the Figure legend. To determine the expression levels of GATA6 could RMAC be included for comparison.

-- An isotype control was used. We have amended Fig. 2E to make this clear and include RMac for comparison.

How does the reduction in GATA6 expression on the IMACZ10 and the genes it regulates impact on function?

Exactly how the level of GATA6 expression determines the identity of LPM and the effect this has on function of LPM is a very interesting question. However, answering this is not easy and while we are currently investigating methods to allow us to experimentally vary the level of GATA6 expression, is not possible to do

these experiments within the timeframe given for the revisions. Hopefully in the future we will be able to address this question specifically.

However, we have taken a broader approach to try and determine whether competition for retinoic acid determines the unique gene expression profile and functionality of IMac^{Z10} including degree of expression of GATA6 and GATA6 regulated genes. First, we performed GSEA on the genes that were differentially expressed between IMac^{Z10} and RMac^{Z10} to identify overlap with those regulated in peritoneal LPM by the RA-activated transcription factors RXR α and RXR β (PMID: 32246014). This revealed that a further fifth of the genes that differentiate IMac^{Z10} from RMac^{Z10} are potentially regulated by RXRAB independently of their effect on GATA6 expression, and this also appeared to be the case with monocyte-derived and established resident cells in naïve mice. Hence, these data provide stronger evidence that differences in the degree of retinoic acid signalling/availability/uptake controls a significant proportion of the different identities of established resident macrophages and incoming monocyte-derived macrophages in steady-state and post-inflammation. We have now included this information in Supplementary Fig. 3g and h.

We then asked whether availability of RA was the limiting factor in the phenotypic conversion of IMac^{Z10} by determining if administration of exogenous all-trans-retinoic acid (ATRA) could rectify the defect in GATA6 expression, and if so, how this would impact the functional difference in their responsiveness to stimulation with LPS. Hence, we adoptively transferred purified IMac^{Z10} and RMac^{Z10} into their native inflamed environment using our standard methodology but combined this with serial intraperitoneal injection of ATRA. However, a pilot experiment in which we gave ATRA in 100% DMSO (as reported in the literature - PMID: 28436955) resulted in overt animal suffering and on the advice of our veterinary team, the experiment was terminated prematurely. In a subsequent experiment ATRA was delivered in oil, which was much better tolerated but unfortunately led to complete loss of donor cells when we culled the animals on day 8 irrespective of whether mice received ATRA or the oil vehicle alone. As these cells had been replaced by recipient Ly6C⁺ monocytes and F4/80^b inflammatory macrophages it seems likely that oil drives disappearance of resident macrophages and inflammatory macrophages recruited during the initial zymosan peritonitis and recruitment of a new wave of inflammatory macrophages. However, we did find that delivery of ATRA significantly increased the number of donor inflammatory macrophages that expressed high and intermediate levels of GATA6. Hence, while this system has not been suitable to address the reviewer's original question, these data demonstrate that retinoic acid availability does limit GATA6 expression by inflammatory macrophages *in vivo*, and so we have now included them in Supplementary Figure 2d-g to make this point.

Do the authors know how these cells differ from RMAC at steady state and during a second inflammatory insult?

-- In Figure 3 we provide evidence that IMac^{Z10} and monocyte-derived LPM recruited in naïve mice exhibit numerous functional and behavioural differences both under steady state conditions and in response to a second inflammatory challenge. Specifically, we demonstrate that these cells undergo heightened proliferation under steady conditions, are poorer at phagocytosis of apoptotic cells *in vivo* and phagocytosis of fluorescently labelled *E coli* *in vitro*, and that they make more IL-10 and less TNF α in response to secondary

challenge with LPS *in vitro* and *in vivo*. We do not yet have the tools to specifically deplete IMac post-inflammation resolution and while we appreciate we have not exhaustively studied the response of these cells to different secondary inflammatory insults, we feel that sufficient evidence is presented to make the point that these cells do differ functionally to RMac and that more in-depth analysis of the functional differences of these cells in difference disease models is better suited to future studies.

It is interesting that following a second insult it is LPM-of recent monocyte origin that proliferate more, measured by ki67. Had the authors considered using EdU to measure proliferation and define when the burst of proliferation occurs or pHistoneH3 as Davies et al EJI 2011 have previously performed? Also, could the intensity of the PKH26 be used to define the extent of proliferation?

-- We have previously shown that LPM of recent monocyte origin proliferate more than established LPM in naïve mice using Ki67 expression and BrdU incorporation, and consistent with this, a subset of LPM can be identified that express higher levels of Ki67 and have a greater history of proliferation as determined by more rapid loss of GFP-labelled histone2B (PMID: 2729202). The greater expansion of IMac^{Z10} in macrophage-depleted mice provides a similar functional measurement of the elevated proliferative capacity of these cells (Figure 2b). Combined, these results give confidence that Ki67 can be used in peritoneal macrophages to accurately identify cycling cells. However, we thank the reviewer for suggesting we use the intensity of PKH26-PCL dye to examine the extent of proliferation of transferred cells. As detailed above, we can now use this to provide convincing evidence that the monocyte-derived fraction of RMac proliferate more than the mature cells in the macrophage-deplete environment (new Supplementary Fig4d-i).

Are these monocyte-LPM following inflammation functioning as repair/resolution macrophages? If so it would interesting to know how the inflammatory microenvironment is unable to redirect their function of these highly plastic cells, and could be discussed.

-- Given the less inflammatory profile of cytokine production by monocyte-derived LPM to secondary stimulation with LPS (Figure 3g-j), it is possible that these cells may underlie the long-term dampening of secondary inflammation in the peritoneal cavity that has been reported to occur following zymosan injection (PMID: 25006125). However, as mentioned above we do not yet have the tools to specifically deplete IMac post-inflammation but this is an on-going area of investigation which we hope will allow us to address this question in the future. With regard to the ability of the inflammatory microenvironment to redirect the function of monocyte-derived macrophages, we are not quite sure what the reviewer is asking here. We feel we have provided convincing evidence to demonstrate that the aberrant state of activation of monocyte-derived LPM is largely due to an inability of these cells to compete with incumbent macrophages for 'niche' signals and that this seems likely to be the case whether monocytes are recruited into an inflamed or naïve cavity, and we note that Reviewer 1 found we had made this case convincingly. However, if the reviewer is questioning why IMac^{Z10} and RMac^{Z10} respond differently *in vivo* to a secondary inflammatory challenge, such as LPS injection (as we show in Figure 3j), and why the secondary inflammatory environment is unable to redirect the function of RMac^{Z10} and IMac^{Z10} to a single coalesced state, it is relatively well established that the response of macrophages to inflammatory challenge is determined by their transcriptional program at the point of activation (e.g. expression of receptors, signaling proteins) as well as the state of the chromatin at the point of

activation (PMID: 27102489). As our Discussion is already extensive and this was not a resolute request, we decided not to include further discussion on this point.

It is no great surprise that the magnitude of inflammation impacts on the inflammatory response. However, it is unexpected how this influences the post inflammation macrophage pool long-term. Unless does inflammation truly vanish with such a large dose of zymosan? Is it possible to confirm all the zymosan particles have been cleared? Alternatively, is the peritoneal cavity in a perpetual state of chronic inflammation?

Re-analysis of our flowcytometry data suggests that events consistent with zymosan particles (SFigure 6g) can be detected in the peritoneal lavage at day 3 post high-dose treatment, but that these are no longer detectable at week 8 (SFigure 6h), suggesting free zymosan has been cleared by this time. However, it is difficult to provide definitive proof that all reservoirs of zymosan or zymosan-derived material have been eliminated from these mice. We have and acknowledged the limitations of this analysis in our Discussion.

To address the question of whether inflammation truly vanishes with such a large dose of inflammation, we went back to our original adoptive transfer experiments to reassess neutrophil numbers and T-cell activation state as markers of chronic on-going inflammation. Surprisingly, this revealed that high dose zymosan led to persistent loss of naïve CD62L⁺ T cells and a low but detectable number of neutrophils at week 8, indicative of chronic low-grade inflammation. We have included this data as Supplementary Figure 6d,e,f and adjusted the text in the Results and Discussion to reflect this.

Although we cannot definitively say why severe inflammation leads to long-term alteration in the composition and activation of peritoneal cavity immune cells, we have briefly discussed potential explanations within the Discussion. However, we feel that more detailed examination of the mechanism underlying this is beyond the scope of the current manuscript, particularly as the main reason for including the high-dose model was to demonstrate that the recruited macrophages more rapidly acquire key features of resident macrophages than those recruited following mild inflammation consistent with the absence of competition from established resident cells following severe but not mild inflammation.

The replacement of LPM following Z1000 and their survival/development into GATA6 positive macrophages is interesting yet these cells don't express CXCL13 like IMACZ10, leading to an altered homeostasis of resident B cells. A final question, maybe for future studies could administration of CXCL13 rescue B cell homeostasis.

-- We agree with the reviewer that this is a very interesting area for future exploration and thank them for their suggestion. Given that the accumulation of peritoneal B1 cells occurs only slowly with age (Figure 4k,l), we had been examining methods to induce continual long-term over-expression of *Cxcl13* by IMac^{Z1000} in order to increase the likelihood of rescuing this process to a measurable degree. However, following the reviewer's suggestion we will investigate the possibility of continual delivery of CXCL13 by peritoneal implantation of long-duration slow release osmotic pumps.

Typo

1.Line 272 Z missing from RMAC10

-- We have corrected this error.

In addition to the above amendments, we spotted a small number of errors and omissions in our citations which we have corrected. Also, we have trimmed our original extensive Discussion to create space for the additional text that has now been included.

We look forward to your consideration.

Stephen Jenkins

Centre for Inflammation Research,
Queens Medical Research Institute,
University of Edinburgh

REVIEWERS' COMMENTS

Reviewer #2 (Remarks to the Author):

Thank you for addressing all of my questions.